# Random genetic drift sets an upper limit on mRNA splicing accuracy in metazoans

Florian Bénitière, Anamaria Necsulea, Laurent Duret*

Laboratoire de Biometrie et Biologie Evolutive, CNRS, Universite Lyon 1, Villeurbanne, France

**Abstract** Most eukaryotic genes undergo alternative splicing (AS), but the overall functional significance of this process remains a controversial issue. It has been noticed that the complexity of organisms (assayed by the number of distinct cell types) correlates positively with their genome-wide AS rate. This has been interpreted as evidence that AS plays an important role in adaptive evolution by increasing the functional repertoires of genomes. However, this observation also fits with a totally opposite interpretation: given that 'complex' organisms tend to have small effective population sizes ($Ne$), they are expected to be more affected by genetic drift, and hence more prone to accumulate deleterious mutations that decrease splicing accuracy. Thus, according to this 'drift barrier' theory, the elevated AS rate in complex organisms might simply result from a higher splicing error rate. To test this hypothesis, we analyzed 3496 transcriptome sequencing samples to quantify AS in 53 metazoan species spanning a wide range of $Ne$ values. Our results show a negative correlation between $Ne$ proxies and the genome-wide AS rates among species, consistent with the drift barrier hypothesis. This pattern is dominated by low abundance isoforms, which represent the vast majority of the splice variant repertoire. We show that these low abundance isoforms are depleted in functional AS events, and most likely correspond to errors. Conversely, the AS rate of abundant isoforms, which are relatively enriched in functional AS events, tends to be lower in more complex species. All these observations are consistent with the hypothesis that variation in AS rates across metazoans reflects the limits set by drift on the capacity of selection to prevent gene expression errors.

## eLife assessment

This **fundamental** study evaluates the evolutionary significance of variations in the accuracy of the intron-splicing process across vertebrates and insects. Using a powerful combination of comparative and population genomics approaches, the authors present **convincing** evidence that higher rates of alternative splicing tend to be observed in species with lower effective population size, a key prediction of the drift-barrier hypothesis. The analysis is carefully conducted and has broad implications beyond the studied species. As such, it will strongly appeal to anyone interested in the evolution of genome architecture and the optimisation of genetic systems.

## Introduction

Eukaryotic protein-coding genes are interrupted by introns, which have to be excised from the primary transcript to produce functional mRNAs that can be translated into proteins. The removal of introns from primary transcripts can lead to the production of diverse mRNAs, via the differential use of splice sites. This process of alternative splicing (AS) is widespread in eukaryotes (**Chen et al., 2014**), but its 'raison d'être' (adaptive or not) remains elusive. Numerous studies have shown that some AS events are functional, that is that they play a beneficial role for the fitness of organisms, either by allowing the production of distinct protein isoforms (**Graveley, 2001**) or by regulating

*For correspondence: Laurent.Duret@univ-lyon1.fr

gene expression post-transcriptionally (*McGlincy and Smith, 2008*; *Hamid and Makeyev, 2014*). However, other AS events are undoubtedly not functional. Like any biological machinery, the spliceosome occasionally makes errors, leading to the production of aberrant mRNAs, which represent a waste of resources and are therefore deleterious for the fitness of the organisms (*Hsu and Hertel, 2009*; *Gout et al., 2013*). The splicing error rate at a given intron is expected to depend both on the efficiency of the spliceosome and on the intrinsic quality of its splice signals. The information required in cis for the removal of each intron resides in 20–40 nucleotide sites, located within the intron or its flanking exons (*Lynch, 2006*). Besides the two splice sites that are essential for the splicing reaction (almost always GT for the donor and AG for the acceptor), all other signals tolerate some sequence flexibility. Population genetics principles state that the ability of selection to promote beneficial mutations or eliminate deleterious mutations depends on the intensity of selection (s) relative to the power of random genetic drift (defined by the effective population size, $Ne$): if the selection coefficient is sufficiently weak relative to drift ($|N_e s| \ll 1$), alleles behave as if they are effectively neutral. Thus, random drift sets an upper limit on the capacity of selection to prevent the fixation of alleles that are sub-optimal (*Kimura et al., 1963*; *Ohta, 1973*). This so-called 'drift barrier' (*Lynch, 2007*) is expected to affect the efficiency of all cellular processes, including splicing. Hence, species with low $Ne$ should be more prone to make splicing errors than species with high $Ne$.

The extent to which AS events correspond to functional isoforms or to errors is a contentious issue (*Bhuiyan et al., 2018*; *Tress et al., 2017b*; *Blencowe, 2017*; *Tress et al., 2017a*). In humans, the set of transcripts produced by a given gene generally consists of one major transcript (the 'major isoform'), which encodes a functional protein, and of multiple minor isoforms (splice variants), present in relatively low abundance, and whose coding sequence is frequently interrupted by premature termination codons (PTCs) (*Tress et al., 2017a*; *Gonzàlez-Porta et al., 2013*). Ultimately, less than 1% of human splice variants lead to the production of a detectable amount of protein (*Abascal et al., 2015*). Furthermore, comparison with closely related species showed that AS patterns evolve very rapidly (*Barbosa-Morais et al., 2012*; *Merkin et al., 2012*) and that alternative splice sites present little evidence of selective constraints (*Pickrell et al., 2010*). All these observations are consistent with the hypothesis that a vast majority of splice variants observed in human transcriptomes simply correspond to erroneous transcripts (*Pickrell et al., 2010*). However, some authors argue that a large fraction of AS events might in fact contribute to regulating gene expression. Indeed, PTC-containing splice variants are recognized and degraded by the non-sense mediated decay (NMD) machinery. Thus, AS can be coupled with NMD to modulate gene expression at the post-transcriptional level (*McGlincy and Smith, 2008*; *Hamid and Makeyev, 2014*). This AS-NMD regulatory process does not involve the production of proteins and does not necessarily imply strong evolutionary constraints on splice sites. Thus, based on these observations, it is difficult to firmly refute selectionist or non-adaptive models.

The analysis of transcriptomes from various eukaryotic species showed substantial variation in AS rates across lineages, with the highest rate in primates (*Barbosa-Morais et al., 2012*; *Chen et al., 2014*; *Mazin et al., 2021*). Interestingly, the genome-wide average AS level was found to correlate positively with the complexity of organisms (approximated by the number of cell types; *Chen et al., 2014*). This correlation was considered as evidence that AS contributed to the evolution of complex organisms by increasing the functional repertoire of their genomes (*Chen et al., 2014*). This pattern is often presented as an argument supporting the importance of AS in adaptation (*Verta and Jacobs, 2022*; *Singh and Ahi, 2022*; *Wright et al., 2022*). However, this correlation is also compatible with a totally opposite hypothesis. Indeed, eukaryotic species with the highest level of complexity correspond to multi-cellular organisms with relatively large body size, which tend to have small effective population sizes ($Ne$) (*Lynch and Conery, 2003*; *Figuet et al., 2016*). Thus, the higher AS rate observed in 'complex' organisms might simply reflect an increased rate of splicing errors, resulting from the effect of the drift barrier on the quality of splice signals (*Bush et al., 2017*).

To assess this hypothesis and evaluate the impact of genetic drift on alternative splicing patterns, we quantified AS rates in 53 metazoan species, covering a wide range of $Ne$ values, and for which high-depth transcriptome sequencing data were available. We show that the genome-wide average AS rate correlates negatively with $Ne$, in agreement with the drift barrier hypothesis. This pattern is mainly driven by low abundance isoforms, which represent the vast majority of splice variants and most likely correspond to errors. Conversely, the AS rate of abundant splice variants, which are enriched

in functional AS events, show the opposite trend. These results support the hypothesis that the drift barrier sets an upper limit on the capacity of selection to minimize splicing errors.

## Results

### Genomic and transcriptomic data collection

To analyze variation in AS rates across metazoans, we examined a collection of 69 species for which transcriptome sequencing (RNA-seq) data, genome assemblies, and gene annotations were available in public databases. We focused on vertebrates and insects, the two metazoan clades that were the best represented in public databases when we initiated this project. To be able to compare average AS rates across species, we needed to control for several possible sources of biases. First, given that AS rates vary across genes (*Saudemont et al., 2017*), we had to analyze a common set of orthologous genes. For this purpose, we extracted from the BUSCO database (*Seppey et al., 2019*) a reference set of single-copy orthologous genes shared across metazoans (N=978 genes), and searched for their homologues in each species in our dataset. We retained for further analyses those species for which at least 80% of the BUSCO metazoan gene set could be identified (N=67 species; see Materials and methods). Second, we had to ensure that RNA-seq read coverage was sufficiently high in each species to detect splicing variants. Indeed, to be able to detect AS at a given intron, it is necessary to analyze a minimal number of sequencing reads encompassing this intron (we used a threshold of N=10 reads). To assess the impact of sequencing depth on AS detection, we conducted a pilot analysis with two species (*Homo sapiens* and *Drosophila melanogaster*) for which hundreds of RNA-seq samples are available. This analysis (detailed in *Figure 2—figure supplement 1*) revealed that AS rate estimates are very noisy when sequencing depth is limited, but that they converge when sequencing is high enough. We therefore kept for further analysis those species for which the median read coverage across exonic regions of BUSCO genes was above 200 (*Figure 2—figure supplement 1*). Our final dataset thus consisted of 53 species (15 vertebrates and 38 insects; *Figure 1A*), and of 3496 RNA-seq samples (66 *per* species on average). In these species, the number of analyzable annotated introns (i.e. encompassed by at least 10 reads) among BUSCO genes ranges from 2032 to 10,981 (which represents 88.6% to 99.6% of their annotated introns; *Figure 1—source data 1*). It should be noted that analyzed samples originate from diverse sources; however, they are very homogenous in terms of sequencing technology (99% of RNA-seq samples sequenced with Illumina platforms; refer to Data10-supp.tab in the Zenodo data repository).

### Proxies for the effective population size (*Ne*)

Effective population sizes (*Ne*) can in principle be inferred from levels of genetic polymorphism. However, population genetics data are lacking for most of the species in our dataset. We therefore used two life history traits that were previously proposed as proxies of *Ne* in metazoans (*Waples, 2016*; *Romiguier and Weyna, 2020*; *Figuet et al., 2016*): body length and longevity (Materials and methods; *Figure 1—source data 2*). An additional proxy for *Ne* can be obtained by studying the intensity of purifying selection acting on protein sequences, through the *dN/dS* ratio (*Kryazhimskiy and Plotkin, 2008*). To evaluate this ratio, we aligned 922 BUSCO genes, reconstructed the phylogenetic tree of the 53 species (*Figure 1A*) and computed the *dN/dS* ratio along each terminal branch (Materials and methods).

We note that these three proxies provide 'inverse' estimates of *Ne*, meaning that species with high longevity, large body length and/or elevated *dN/dS* values tend to have low *Ne* values. As expected, these different proxies of *Ne* are positively correlated with each other (p $<1 \times 10^{-3}$, *Figure 1B and C*). We note however that these correlations are not very strong. It thus seems likely that none of these proxies provides a perfect estimate of *Ne*. To take phylogenetic inertia into account, all cross-species correlations presented here were computed using Phylogenetic Generalized Least Squared (PGLS) regression (*Freckleton et al., 2002*).

### Alternative splicing rates are negatively correlated with *Ne* proxies

To quantify AS rates, we mapped RNA-seq data of each species on the corresponding reference genome assembly. We detected sequencing reads indicative of a splicing event (hereafter termed 'spliced reads'), and inferred the corresponding intron boundaries. We were thus able to validate

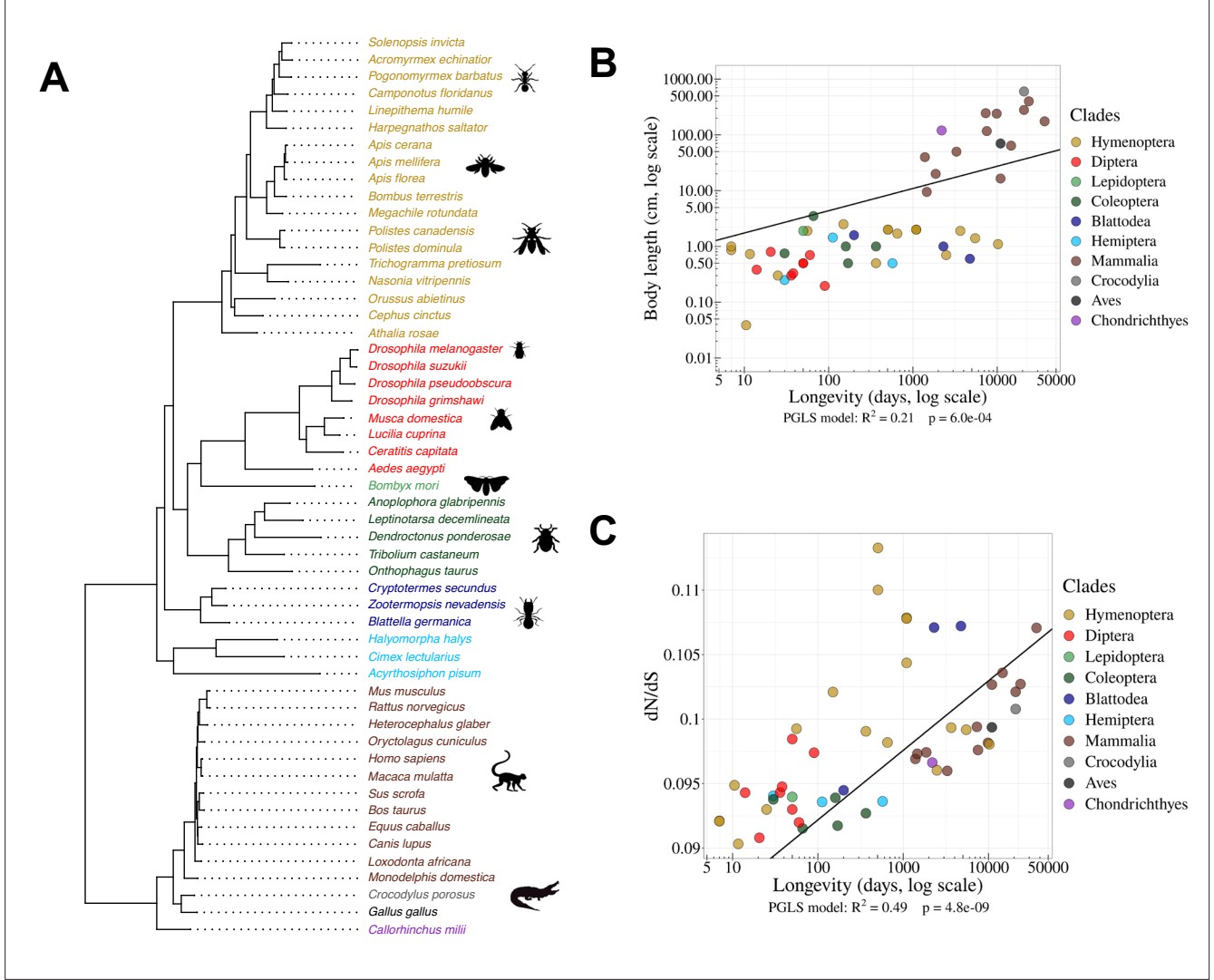

**Figure 1.** Species phylogeny and *Ne* proxies. (**A**) Phylogenetic tree of the 53 studied species (15 vertebrates and 38 insects). (**B**) Relationship between body length (cm, log scale) and longevity (days, log scale) of the organism. Each dot represents one species (colored by clade, as in the species tree in panel A). (**C**) Relationship between longevity (days, log scale) and the *dN/dS* ratio on terminal branches of the phylogenetic tree (Materials and methods). (**B,C**) PGLS stands for Phylogenetic Generalized Least Squared regression, which takes into account phylogenetic inertia (Materials and methods).

The online version of this article includes the following source data for figure 1:

**Source data 1.** Summary of the main features of the samples analyzed in this study.

**Source data 2.** Longevity and body lenth across the 53 metazoans studied.

the coordinates of annotated introns and to detect new introns, not present in the annotations. For each intron detected in RNA-seq data, we counted the number of spliced reads matching with its two boundaries ($N_s$) or sharing only one of its boundaries ($N_a$), as well as the number of unspliced reads covering its boundaries ($N_u$) (*Figure 2A*). We then computed the relative abundance of this spliced isoform compared to other transcripts with alternative splice boundaries (RAS = $\frac{N_s}{N_s+N_a}$) or compared to unspliced transcripts (RANS = $\frac{N_s}{N_s+\frac{N_u}{2}}$).

To limit measurement noise, we only considered introns for which both RAS and RANS could be computed based on at least 10 reads (Materials nd methods). In all species, both RAS and RANS metrics show clearly bimodal distributions (*Figure 2B and C*): the first peak (mode < 5%) corresponds to 'minor-isoform introns', whose splicing occurs only in a minority of transcripts of a given

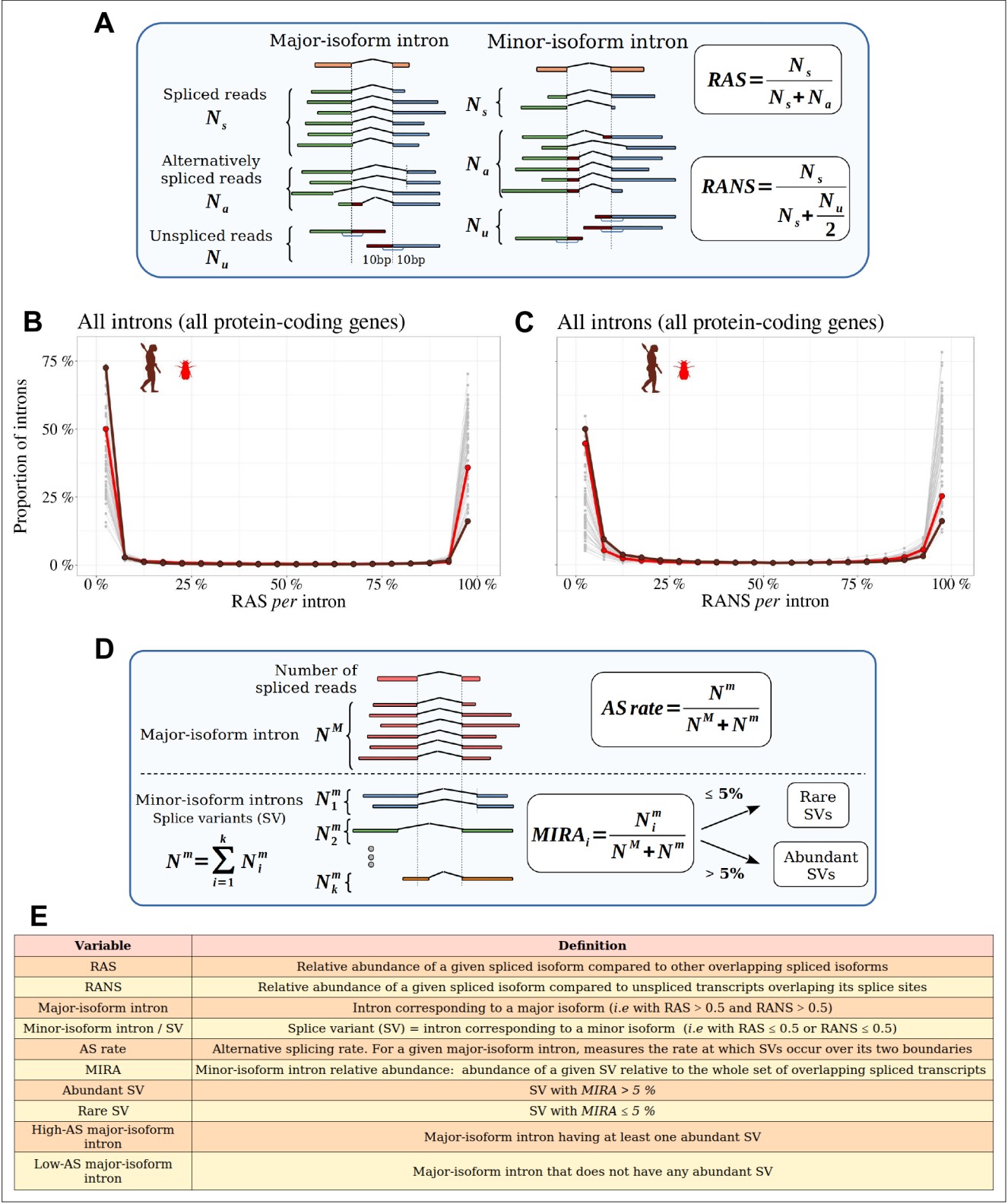

**Figure 2.** Distinguishing major and minor-isoform introns and measuring the rate of alternative splicing. (**A**) Definition of the variables used to compute the relative abundance of a spliced isoform compared to other transcripts with alternative splice boundaries (RAS) or compared to unspliced transcripts (RANS): $N_s$: number of spliced reads corresponding to the precise excision of the focal intron; $N_a$: number of reads corresponding to alternative splice variants relative to this intron (i.e. sharing only one of the two intron boundaries); $N_u$: number of unspliced reads, co-linear with the genomic sequence. (**B,C**) Histograms representing the distribution of RAS and RANS values (divided into 5% bins), for protein-coding gene introns. Each line represents

*Figure 2 continued on next page*

*Figure 2 continued*

one species. Two representative species are colored: *Drosophila melanogaster* (red), *Homo sapiens* (brown). (**D**) Description of the variables used to compute the AS rate of a given a major-isoform intron, and the 'minor-isoform intron relative abundance' (MIRA) of each of its splice variants (SVs): $N^M$: number of spliced reads corresponding to the excision of the major-isoform intron; $N_i^m$: number of spliced reads corresponding to the excision of a minor-isoform intron (i); $N^m$: total number of spliced reads corresponding to the excision of minor-isoform introns. (**E**) Definitions of the main variables used in this study.

The online version of this article includes the following figure supplement(s) for figure 2:

**Figure supplement 1.** Transcriptome sequencing depth affects intron detection power and AS rate estimates.

**Figure supplement 2.** The power to detect AS events is positively correlated with transcriptome sequencing depth.

**Figure supplement 3.** Description of the bioinformatic analyses pipeline.

gene, whereas the second one (mode > 95%) corresponds to the introns of major isoforms. It has been previously shown that in humans, for most genes, one single transcript largely dominates over other isoforms (*Tress et al., 2017a*; *Gonzàlez-Porta et al., 2013*). Our observations indicate that this pattern is generalized across metazoans. For the rest of our analyses, we computed the rate of alternative splicing with respect to introns of the major isoform. We will hereafter use the term 'splice variant' (SV) to refer to those splicing events that are detected in a minority of transcripts (i.e. with RAS ≤ 0.5 or RANS ≤ 0.5; see *Figure 2E* for a definition of the main variables used in this study).

We focused our analyses on major-isoform introns interrupting protein-coding regions (i.e. we excluded introns located within UTRs, Materials and methods). In vertebrates, each BUSCO gene contains on average 8.4 major-isoform introns (*Figure 1—source data 1*). The intron density is more variable among insect clades, ranging from 2.8 major-isoform introns per BUSCO gene in Diptera to 6.1 in Blattodea. As expected, most major-isoform introns have GT/AG splice sites (99.1% on average across species), and only a small fraction have boundaries that do not match the canonical U2-introns splice sites (0.8% GC/AG and 0.1% AT/AC). The fraction of non-canonical splice sites is slightly higher among minor-isoform introns (2.8% GC/AG and 0.3% AT/AC). This might reflect a higher prevalence of U12-type introns but might also be caused by the presence of some false positives in the set of minor-isoform introns. In any case, the difference in splice signal usage between minor and major-isoform introns is small, which indicates that the vast majority of detected minor-isoform introns correspond to bona fide splicing events.

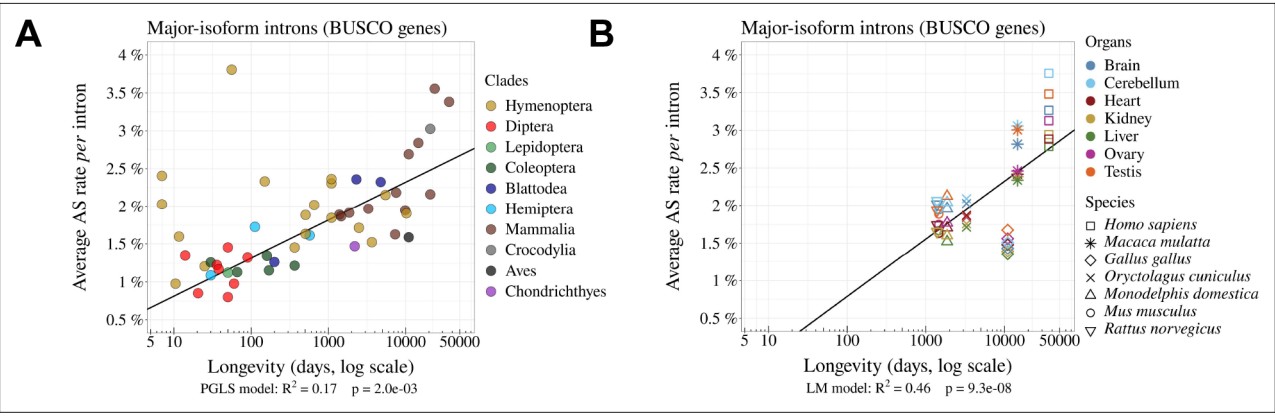

**Figure 3.** The rate of alternative splicing correlates with life history traits across metazoans. (**A**) Relationship between the *per* intron average AS rate of an organism and its longevity (days, log scale). (**B**) Variation in average AS rate across seven organs (brain, cerebellum, heart, liver, kidney, testis, and ovary) among seven vertebrate species (RNA-seq data from *Cardoso-Moreira et al., 2019*). AS rates are computed on major-isoform introns from BUSCO genes (Materials and methods).

The online version of this article includes the following figure supplement(s) for figure 3:

**Figure supplement 1.** Relationship between AS rates and other *Ne* proxies.

**Figure supplement 2.** The rate of alternative splicing correlates with life history traits in both vertebrates and insects.

**Figure supplement 3.** The variation in AS rates between species is not explained by organ differences.

**Figure supplement 4.** The *per*-gene AS rate is negatively correlated with *Ne*.

The proportion of major-isoform introns for which AS has been detected (i.e. with Na>0) ranges from 16.8% to 95.7% depending on the species (*Figure 1—source data 1*). This metric is however not very meaningful because it directly reflects differences in sequencing depth across species (the higher the sequencing effort, the higher the probability to detect a rare SV, *Figure 2—figure supplement 2*). To allow a comparison across taxa, we computed the AS rate of introns, normalized by sequencing depth (AS $= \frac{N^m}{N^M + N^m}$, Materials and methods; *Figure 2D*). The average AS rate for BUSCO genes varies by a factor of 5 among species, from 0.8% in *Drosophila grimshawi* (Diptera) to 3.8% in *Megachile rotundata* (Hymenoptera) (3.4% in humans). Interestingly, the average AS rates of BUSCO gene introns are significantly correlated with the three proxies of *Ne*: species longevity (*Figure 3A*), body length and the *dN/dS* ratio (*Figure 3—figure supplement 1A and B*). These correlations are positive, which implies that AS rates tend to increase when *Ne* decreases. It is noteworthy that despite the fact that these proxies are not strongly correlated with each other (*Figure 1B and C*), they all show similar relationships with AS rates. It should be stressed that these correlations were estimated using the PGLS method to account for phylogenetic inertia (and they remain significant when analyzing insects and vertebrates separately, *Figure 3—figure supplement 2*). Thus, these observations are consistent with the hypothesis that *Ne* has an impact on the evolution of AS rate.

One limitation of our analyses is that we used heterogeneous sources of transcriptomic data. To obtain enough sequencing depth, we combined for each species many RNA-seq samples, irrespective of their origin (whole body, or specific tissues or organs, in adults or embryos, etc.). It is known that genome-wide average AS rates vary according to tissues or developmental stages (*Barbosa-Morais et al., 2012*; *Mazin et al., 2021*), and according to environmental conditions (*John et al., 2021*). To explore how this might have affected our results, we repeated our analyses using a recently published dataset that aimed to compare transcriptomes across seven organs, sampled at several developmental stages in seven species (six mammals, one bird; *Cardoso-Moreira et al., 2019*). In agreement with previous reports (*Mazin et al., 2021*), our analysis of BUSCO genes revealed substantial differences in AS rates among organs, with consistent patterns of variation across species. For instance, in all species, testes and brain tissues show higher AS rates than liver and kidney (*Figure 3B*). However, the variation in AS rate among organs in each species is limited compared to differences between species. Specifically, in an ANOVA analysis performed on the average AS rate across BUSCO gene introns, with the species and the organ of origin as explanatory variables, the species factor explained 89% of the total variance, while the organ factor explained only 9%. Among insects, we found only one species (*Dendroctonus ponderosae*) for which RNA-seq samples were available from multiple tissues. Here again, the variance in AS rate among tissues was limited compared to inter-species variability (*Figure 3—figure supplement 3*). Thus, despite the variability that can be introduced by the heterogeneity of RNA-seq samples, the relationship between AS rate and longevity remains detectable among these seven species (*Figure 3B*).

## Functional vs. non-functional alternative splicing

The negative correlation observed between *Ne* and alternative splicing rates is consistent with the hypothesis that differences in AS rates across species are driven by variation in the rate of splicing errors (drift barrier model). This does not exclude however that functional splicing variants might also contribute to AS rate variation across species. To evaluate this point, we selected a subset of SVs that are enriched in functional AS events. To do this, we reasoned that selective pressure against the waste of resources should maintain splicing errors at a low rate (as low as permitted by the drift barrier), whereas functional SVs are expected to represent a sizeable fraction of the transcripts expressed by a given gene, at least in some specific conditions (cell type, developmental stage…). Thus, functional SVs are expected to be enriched among abundant SVs compared to rare SVs.

To assess this prediction, we analyzed the proportion of SVs that preserve the reading frame according to their abundance relative to the major isoform. For this, we focused on minor-isoform introns that share a boundary with one major-isoform intron and that have their other boundary at less than 30 bp from the major splice site (either in the flanking exon or within the major-isoform intron). We determined whether the distance between the minor-isoform intron boundary and the major-isoform intron boundary was a multiple of 3. We computed the abundance of each minor isoform, relative to the corresponding major isoform, with the following formula: Minor intron relative abundance $\text{MIRA}_i = \frac{N_i^m}{N^M + N^m}$ (see *Figure 2D*).

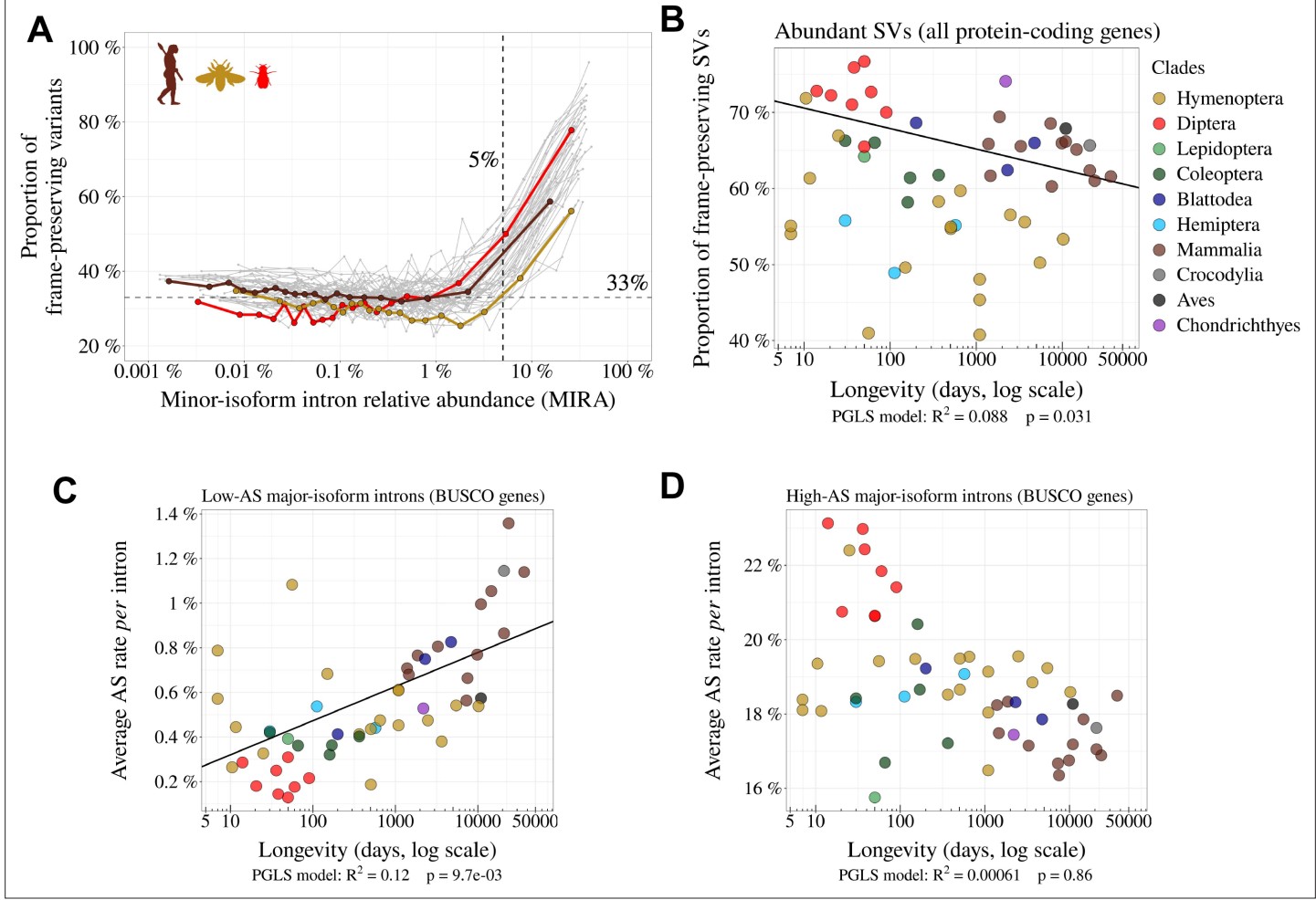

**Figure 4.** Variation in AS rate across metazoans: distinguishing abundant splice variants (enriched in functional variants) from rare splice variants. (**A**) Frame-preserving isoforms are strongly enriched among abundant splice variants (SVs). For each species, SVs were classified into 20 equal-size bins according to their abundance relative to the major isoform (MIRA, see Materials and Methods), and the proportion of frame-preserving SVs was computed for each bin. Each line represents one species. Three representative species are colored: red: *Drosophila melanogaster*, brown: *Homo sapiens*, yellow: *Apis mellifera*. We used a threshold MIRA value of 5% to define 'abundant' vs. 'rare' SVs. (**B**) Proportion of frame-preserving SVs among abundant SVs across metazoans. Each dot represents one species. All annotated protein-coding genes are used in the analysis. (**C,D**) Relationship between the average *per* intron AS rate of an organism and its longevity (days, log scale). Only BUSCO genes are used in the analysis. (**C**) Low-AS major-isoform introns (i.e. major-isoform introns that do not have any abundant SV), (**D**) High-AS major-isoform introns (i.e. major-isoform introns having at least one abundant SV).

The online version of this article includes the following figure supplement(s) for figure 4:

**Figure supplement 1.** Relationship between AS rates and *Ne* proxies, for all major-isoform introns, low-AS major-isoform introns (i.e. major-isoform introns that do not have any abundant spliced variants) and high-AS major-isoform introns (i.e. major-isoform introns having at least one abundant spliced variants).

**Figure supplement 2.** Relationship between the proportion of frame-preserving SVs and *Ne* proxies.

We divided minor-isoform introns into 5% bins according to their MIRA and computed for each bin the proportion of minor-isoform introns that maintain the reading frame of the major isoform (*Figure 4A*). In all species, we observe that this proportion varies according to the abundance of splice variants, with two distinct regimes (*Figure 4A*). First, for MIRA values above 5%, the proportion of frame-preserving variants correlates positively with MIRA, reaching up to 60–70% for the most abundant isoforms. Second, for MIRA values below 1%, the proportion of frame-preserving variants does not covary with MIRA, and fluctuates around 30–40%, close to the random expectation (33%). The excess of frame-preserving variants among the most abundant isoforms implies that a substantial fraction of them is under constraint to encode functional protein isoforms. This fraction varies from

0% for MIRA values below 1%, to 50% for isoforms with the highest MIRA values. It should be noted that these estimates correspond to a lower bound, since it is possible that some frame-shifting splice variants are functional. Nevertheless, these observations clearly indicate that the subset of SVs with MIRA values >5% (hereafter referred to as 'abundant SVs') is strongly enriched in functional isoforms relative to other SVs (MIRA ≤ 5%, hereafter referred to as 'rare SVs'). Of note, the subset of rare SVs represents the vast majority of the SV repertoire (from 62.4% to 96.9% depending on the species; *Figure 1—source data 1*). Thus, the positive correlation between AS rate and longevity reported above (*Figure 3A*) is mainly driven by the set of introns with a low AS rate (*Figure 4C*). Interestingly, introns with high AS rate (enriched in functional SVs) show an opposite trend (*Figure 4D*), and they display a lower proportion of frame-preserving SVs in vertebrates than in dipterans (*Figure 4B*). This is the opposite of what would have been expected if functional SVs were more prevalent in complex organisms.

## Investigating selective pressures on minor splice sites

A complementary approach to assess the functionality of AS events consists in investigating signatures of selective constraints on splice sites. For this, we used polymorphism data from *Drosophila melanogaster* and *Homo sapiens* to measure single-nucleotide polymorphism (SNP) density at major

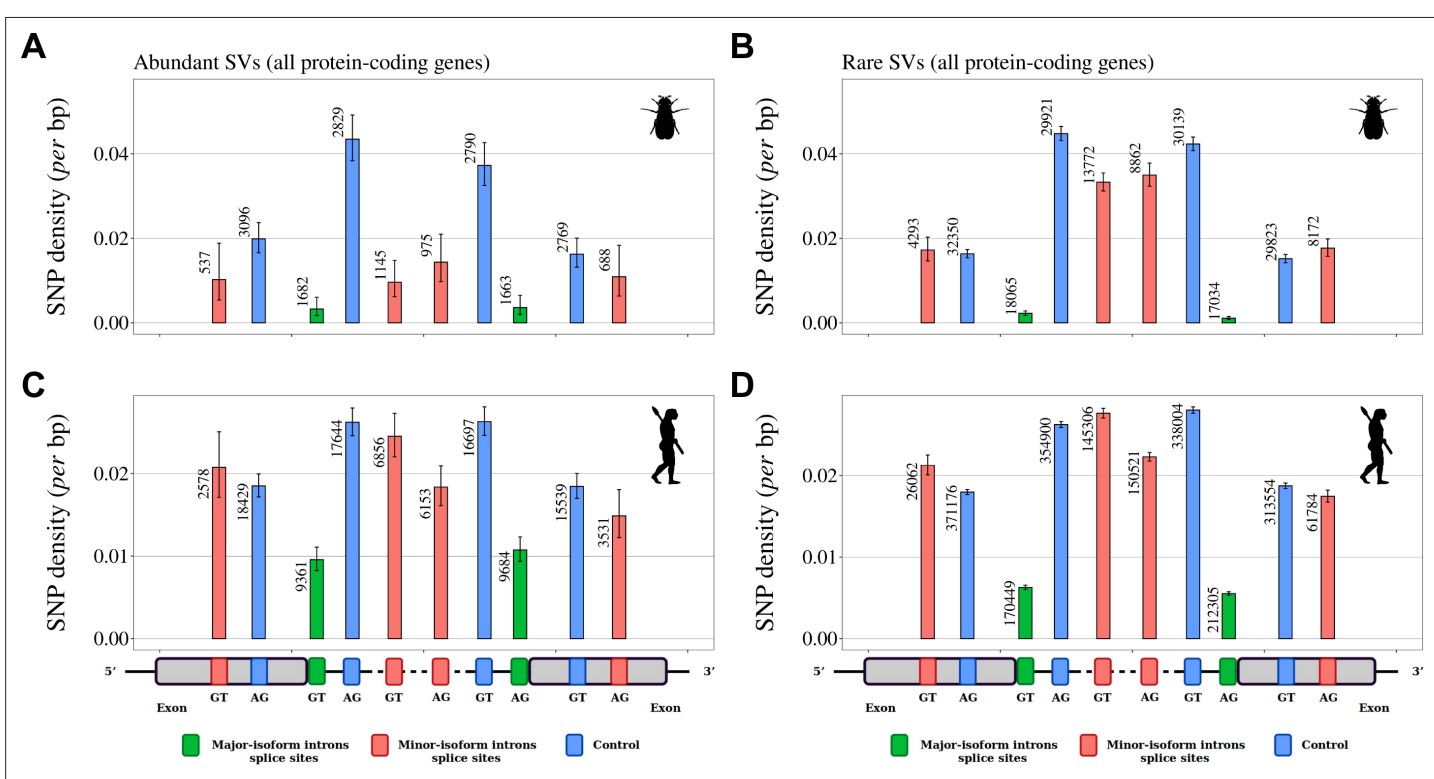

**Figure 5.** Variation in selective constraints on alternative splice signals from rare and abundant SVs. For each minor-isoform intron sharing one boundary with a major-isoform intron, we measured the SNP density at its minor splice site (red), and at the corresponding major splice site (green). We distinguished minor splice sites that are located in an exon or in an intron of the major isoform. As a control (blue), we selected AG or GT dinucleotides that are unlikely to correspond to alternative splice sites, namely: AG dinucleotides located toward the end of the upstream exon or the beginning of the intron (unlikely to correspond to a genuine acceptor site), and GT dinucleotides located toward the beginning of the downstream exon or the end of the intron (unlikely to correspond to a donor site). To increase the sample size, we analyzed data from all annotated protein-coding genes (and not only the BUSCO gene set). The number of sites studied is shown at the top of each bar. Error bars represent the 95% confidence interval of the proportion of polymorphic sites (proportion test). (**A,B**) SNP density in *Drosophila melanogaster* (polymorphism data from 205 inbred lines derived from natural populations, N=3,963,397 SNPs *Huang et al., 2014*; *Mackay et al., 2012*). (**C,D**) SNP density in *Homo sapiens* (polymorphism data from 2504 individuals, N=80,868,061 SNPs *Auton et al., 2015*). We excluded dinucleotides affected by CpG hypermutability (Materials and methods, see *Figure 5—figure supplement 1* for CpG sites). (**A,C**) Abundant SVs (MIRA > 5%). B,D: Rare SVs (MIRA ≤ 5%).

The online version of this article includes the following figure supplement(s) for figure 5:

**Figure supplement 1.** SNP density in human splice signals, for dinucleotides affected by CpG hypermutability.

and minor splice sites, considering separately rare and abundant SVs. We focused on the first two and last two bases of each intron (consensus sequences GT, AG), which represent the most constrained sites within splice signals. We studied minor-isoform introns that share one splice site with a major-isoform intron and we measured SNP density at the corresponding major and minor splice sites. To account for constraints acting on coding regions, we considered separately minor splice sites that were located in an exon or in an intron of the major isoform. As negative controls, we selected AG or GT dinucleotides that were unlikely to correspond to alternative splice sites (*Figure 5*, Materials and methods). Furthermore, for *Homo sapiens* we controlled for the presence of hypermutable CpG dinucleotides (*Tomso and Bell, 2003*; *Figure 5—figure supplement 1*, Materials and methods).

For both species, the lowest SNP density is observed at major splice signals, which reflects the strong selective constraints on these sites (*Figure 5*). In *Drosophila melanogaster*, there is also a strong signature of selection on minor splice signals of abundant SVs: both in introns and in exons, the SNP density at minor splice signals of abundant SVs is much lower than in corresponding controls (from –37% to –74%, *Figure 5A*) and than in minor splice signals of rare SVs (from –38% to –71%, *Figure 5B*). This observation confirms that abundant SVs are strongly enriched in functional variants compared to rare SVs. In *Homo sapiens*, patterns of SNP density showed little evidence of selective constraints on minor splice sites, irrespective of the abundance of SVs (*Figure 5C and D*): minor acceptor splice sites (AG) located within the major-isoform intron show a weak but significant SNP deficit relative to corresponding control sites (p-value $<1 \times 10^{-5}$), but other categories of minor splice sites do not show any sign of selective constraints. The fact that the signature of selection on minor splice signals is much weaker in humans compared to *Drosophila* is indicative of a lower prevalence of functional variants, even among abundant SVs. This observation is therefore in total contradiction with the adaptive hypothesis (more functional alternative splicing in complex organisms).

## The splicing rate of rare SVs is negatively correlated with gene expression levels

The above analyses are consistent with the hypothesis that the vast majority of rare SVs correspond to erroneous transcripts, and that changes in Ne contribute to variation in AS rate across taxa by shifting the selection-mutation-drift balance. If true, then this model predicts that the erroneous AS rate should also vary among genes, according to their expression level. Indeed, it has been shown that the selective pressure on splicing accuracy is stronger on highly expressed genes (*Saudemont et al., 2017*). This reflects the fact that for a given splicing error rate, the waste of resources (both in terms of metabolic cost and of futile mobilization of cellular machineries) increases with gene expression level (*Saudemont et al., 2017*; *Xiong et al., 2017*). Thus, the selection-mutation-drift balance should lead to a negative correlation between gene expression level and the rate of splicing errors. To test this prediction, we focused on low-AS major-isoform introns, *i.e.* introns that are unlikely to have functional SVs. For each species, we considered all major-isoform introns with a sufficient sequencing depth to have a precise measure of their AS rate ($N_s + N_a \geq 100$). The selected subset represents 38.1% to 86.7% of major-isoform introns of each species (median = 70.9%). Introns were then divided into 20 bins of equal size, according to the expression level of the corresponding genes. For each species, we computed the Pearson correlation between the average AS rate and the average expression level across bins. We observed a negative correlation between AS rates and gene expression levels in 52 out of the 53 species (significant with p < 0.05, in 48/53 species; *Figure 6A*; two representative examples are shown in *Figure 6C and D*). This pattern indicates that in almost all metazoan species, genes with a higher expression level have a lower AS rate, consistent with the hypothesis the rate of splicing errors is shaped by the selection-mutation-drift balance. It should be noted that this negative correlation between AS rate and gene expression level is not expected for functional SVs (there is a priori no reason why the AS rate of functional SVs should be higher in weakly expressed genes than in highly expressed genes). Interestingly, when we performed this analysis on all introns (including those with abundant SVs, which are enriched in functional variants), then most species (31/53) still showed a negative correlation between AS rate and gene expression level (*Figure 6B*), but some species, such as *Drosophila melanogaster* showed the opposite pattern (*Figure 6—figure supplement 1*). This probably reflects that fact that, in those species, functional AS events make a significant contribution to the genome-wide average AS rate.

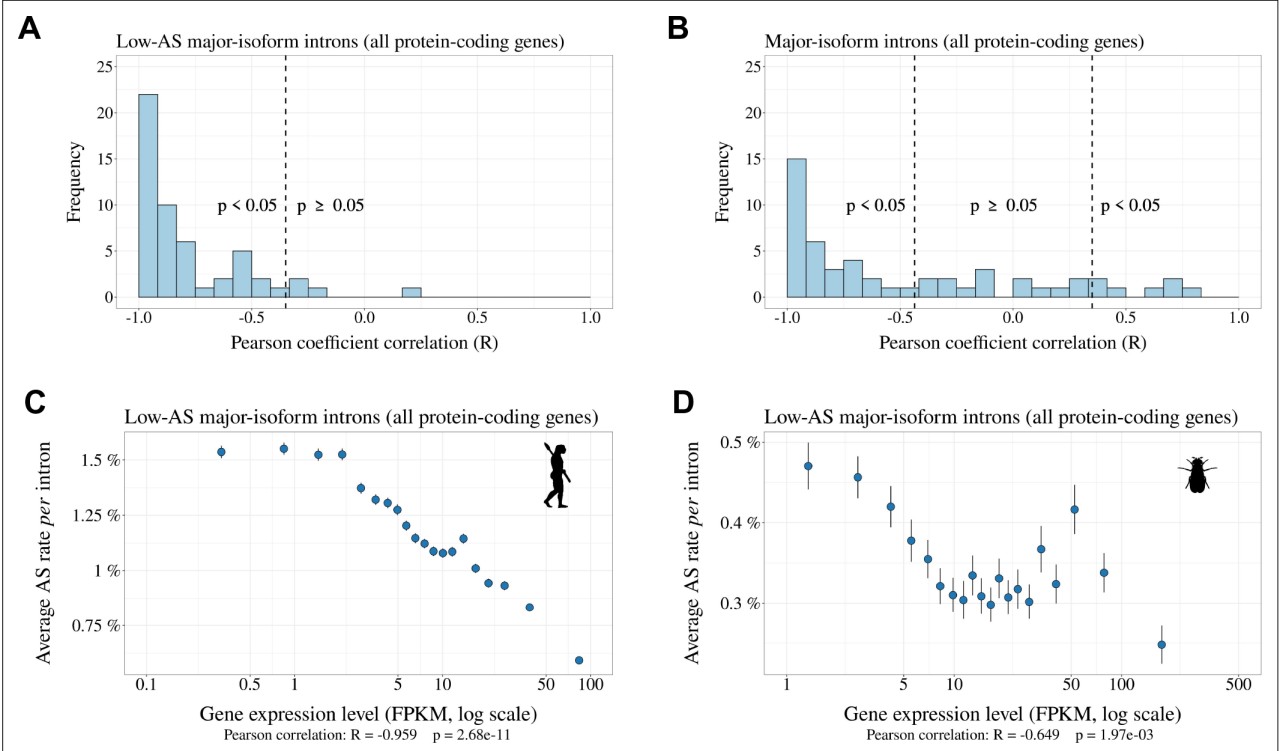

**Figure 6.** Relationship between AS rate and gene expression level. For each species, we selected major-isoform introns with a sufficient sequencing depth to have a precise measure of their AS rate ($N_s + N_a \geq 100$). We divided major-isoform introns into 5% bins according to their gene expression level and computed the correlation between the average AS rate and median expression level across the 20 bins. To increase sample size, these analyses were based on all annotated protein-coding genes (and not only the BUSCO gene set). (**A**) Distribution of Pearson correlation coefficients (R) between the AS rate and expression level observed in the 53 metazoans. The vertical dashed lines indicates the thresholds under and above which correlations are significant (i.e. p-value < 0.05). (**B**): Distribution of Pearson correlation coefficients computed on the subsets of low-AS major-isoform introns (i.e. after excluding major-isoform introns with abundant SVs). (**C,D**) Two representative species illustrating the negative relation between the average AS rate of low-AS major-isoform introns and the expression level of their gene. Error bars represent the standard error of the mean. (**C**) N=127,599 low-AS major-isoform introns from *Homo sapiens*, (**D**) N=31,357 low-AS major-isoform introns from *Drosophila melanogaster*.

The online version of this article includes the following figure supplement(s) for figure 6:

**Figure supplement 1.** Correlations between gene expression levels and AS rates differ among species.

## Discussion

To investigate the factors that drive variation in AS rates across species, we analyzed publicly available RNA-seq data across a large set of 53 species, from diverse metazoan clades, covering a wide range of *Ne* values. To facilitate comparisons across species, we sought to limit the impact of the among-gene variance in AS rates. For this, we primarily based our analyses on a common set of nearly 1000 orthologous protein-coding genes (BUSCO gene set). We focused our study on introns located within protein-coding regions, because introns from UTRs or lncRNAs are expected to be subject to different functional constraints. We measured AS rates on introns corresponding to a major isoform. When sequencing depth is limited, the set of introns for which AS can be quantified is biased toward the most highly expressed genes. To avoid this bias, we restricted our study to species for which the median sequencing depth of BUSCO exons was above 200. With this setting, on average 96.9% of BUSCO annotated introns could be analyzed in each species (*Figure 1—source data 1*).

We observed a fivefold variation in the average AS rate of BUSCO introns across species from 0.8% in *Drosophila grimshawi* (Diptera) to 3.8% in *Megachile rotundata* (Hymenoptera)(*Figure 3A*). In agreement with previous work, we observed that AS rates tend to be high in vertebrates (average = 2.3%), and notably in primates (average = 3.1%) (*Barbosa-Morais et al., 2012*; *Chen et al., 2014*; *Mazin et al., 2021*). This observation was previously interpreted as an evidence that AS played an important role in the diversification of the functional repertoire necessary for the development of more complex organisms (*Chen et al., 2014*). However, this pattern is also compatible with the

hypothesis that variation in AS rates across species result from differences in splicing error rates, which are expected to be higher in species with low *Ne* (*Bush et al., 2017*). Indeed, consistent with this drift barrier hypothesis, we observed significant correlations between AS rates and proxies of *Ne* (*Figure 3B*, *Figure 3—figure supplement 1A and B*).

In their original study, *Chen et al., 2014* investigated the hypothesis that variation in AS rates across taxa might be driven by variation in *Ne*. For this, they focused on 12 species, for which they had measured levels of polymorphism at silent sites ($\pi$). They found that the correlation between AS rate and the number of cell types (proxy for organismal complexity) remained significant after controlling for $\pi$. They therefore concluded that the association between the cellular diversity and alternative splicing was not a by-product of reduced effective population sizes among more complex species. This conclusion was however based on a very small sample of species. More importantly, it assumed that $\pi$ could be taken as a proxy for *Ne*. At mutation-drift equilibrium, $\pi$ is expected to be proportional to *Ne*u (where u is the mutation rate *per* bp *per* generation). Thus, if u is constant across taxa, $\pi$ can be used to estimate variation in *Ne*. However, the dataset analyzed by *Chen et al., 2014* included very diverse eukaryotic species, with mutation rates ranging from 1.7 x$10^{10}$ mutation *per* bp *per* generation in budding yeast, to 1.1 x$10^8$ mutation *per* bp *per* generation in humans (*Lynch et al., 2016*). Hence, at this evolutionary scale, variation in *Ne* cannot be directly inferred from $\pi$ without accounting for variation in u. Moreover, the drift barrier hypothesis states that the AS rate of a species should reflect the genome-wide burden of slightly deleterious substitutions, which is expected to depend on the intensity of drift over long evolutionary times (i.e. long-term *Ne*). Conversely, $\pi$ reflects *Ne* over a short period of time (of the order of *Ne* generations), and can be strongly affected by recent population bottlenecks (too recent to have substantially impacted the genome-wide deleterious substitution load). The drift barrier hypothesis therefore predicts that the splicing error rate should correlate more strongly with proxies of long-term *Ne* (such as *dN/dS*, life history traits, or organismal complexity) than with $\pi$. The fact that AS rates remained significantly correlated to cellular diversity after controlling for $\pi$(*Chen et al., 2014*) is therefore not a conclusive argument against the drift barrier hypothesis.

To contrast the two models (drift barrier vs diversification of the functional repertoire in complex organisms), we sought to distinguish functional splice isoforms from erroneous splicing events. Based on the assumption that splicing errors should occur at a low frequency, we split major-isoform introns into two categories, those with abundant SVs (MIRA > 5%), and those without (MIRA $\leq$ 5%). Rare SVs represent the vast majority of the repertoire of splicing isoforms detected in a given transcriptome (from 62.4% to 96.9% according to the species; *Figure 1—source data 1*). Two lines of evidence indicate that the small subset of abundant isoforms is strongly enriched in functional transcripts relative to other SVs. First, we observed that in all species, the proportion of SVs that preserve the reading frame is much higher among abundant SVs than among rare SVs (*Figure 4A*). Second, the analysis of polymorphism data in *Drosophila* indicates that the average level of purifying selection on alternative splice sites is much stronger for abundant than rare SVs (*Figure 5A and B*).

If variation in AS rate across species had been driven by a higher prevalence of functional SVs in more complex organisms, one would have expected the proportion of frame-preserving SVs to be stronger in vertebrates than in insects, in particular for the set of introns with high AS rate (i.e. enriched in functional SVs). On the contrary, the highest proportion of frame-preserving SVs is observed in dipterans (*Figure 4B*). In fact, the overall higher AS rate of vertebrates (*Figure 3A*) is driven by the set of introns with a low AS rate (*Figure 4C*), that is the set of introns in which the prevalence of functional SVs is the lowest. On the contrary, among the set of introns with high AS rate, vertebrates have lower AS rates than insects (*Figure 4D*).

These observations are difficult to reconcile with the hypothesis that the higher AS rate in vertebrates results from a higher rate of functional AS. Conversely, these observations fit very well with a model where variation in AS rate across species is entirely driven by variation in the efficacy of selection against splicing errors. To illustrate this model, let us consider three hypothetical species with different *Ne*, in which a small fraction of major-isoform introns (say 5%) is subject to functional alternative splicing. Let us consider that the distribution of AS rates of functional splicing variants is the same for all species (i.e. independent of *Ne*), with a mean of 25% (and a standard deviation of 5%). In addition, we assume that all major-isoform introns are potentially affected by splicing errors, with a mean error rate ranging from 0.2% in species of high *Ne* to 1.2% in species of low *Ne*, owing to the drift barrier effect (these parameters were set to match approximately the AS rates observed

in empirical data for rare SVs). The distributions of AS rate given by this model are presented in *Figure 7A*: rare SVs (MIRA ≤ 5%) essentially correspond to splicing errors, while abundant SVs (MIRA > 5%) correspond to a mixture of functional and spurious variants, whose relative proportion depend on *Ne* (*Figure 7B*). This simple model makes predictions that match with our observations: we noted a positive correlation between AS rate and longevity (i.e. a negative correlation with *Ne*) for the set of low-AS major-isoform introns (*Figure 4C*), but an opposite trend for high-AS major-isoform introns (*Figure 4D*), as predicted by the model (*Figure 7D and E*). Given that high-AS major-isoform introns represent only a small fraction of major-isoform introns, this model predicts that, overall, AS rates correlate negatively with *Ne* (*Figure 7*), as observed in empirical data (*Figure 3A*, *Figure 3—figure supplement 1*).

It should be noted that the BUSCO dataset corresponds to genes that are strongly conserved across species, often highly expressed, and hence might not be representative of the entire genome. Notably, AS rates are on average lower in the BUSCO gene set than in other genes, even after accounting for their expression level (*Figure 6—figure supplement 1*). However, results remained qualitatively unchanged when we repeated our analyses on the whole set of annotated protein-coding genes for each species: correlations between AS rates and *Ne* proxies are slightly weaker than on the BUSCO subset, but remain significant (*Figure 4—figure supplement 1*).

The model also predicts that the proportion of functional SVs among high-AS major-isoform introns should vary with *Ne* (*Figure 7C*). To assess this point, we measured in each species the enrichment in reading frame-preserving events among abundant SVs compared to rare SVs. As predicted, this estimate of the prevalence of functional SVs tends to decrease with decreasing *Ne* proxies (e.g. *Figure 4B*, where *Ne* is approximated by longevity). However, these correlations are weak, marginally significant after accounting for phylogenetic inertia with only two of the three *Ne* proxies, and not robust to multiple testing issues (*Figure 4—figure supplement 2*). Thus, *Ne* does not appear to be a strong predictor of the prevalence of functional SVs among high-AS major-isoform introns.

According to the drift-barrier model, the level of splicing errors is expected to decrease with increasing selective pressure. In all above analyses, we considered AS rates measured *per* intron, and not *per* gene. Yet, the trait under selection is the *per*-gene error rate, which depends not only on the error rate *per* intron, but also on the number of introns *per* gene. Given that intron density varies widely across clades (from 2.8 introns *per* gene in diptera to 8.4 introns *per* gene in vertebrates; ), the correlations reported above between AS rates and *Ne* may undervalue the predictive power of the drift-barrier model. The RNA-seq datasets that we analyzed consist of short-read sequences, which do not allow a direct quantification of the *per*-gene AS rate. We therefore indirectly estimated the *per*-gene AS rate in each species, based on the *per*-intron AS rate and on the number of introns *per* gene (Materials and methods). Interestingly, as predicted by the drift-barrier model, *Ne* proxies correlate more strongly with this estimate of the *per*-gene AS than with the *per*-intron AS rates (*Figure 3—figure supplement 4*).

One other important prediction of the drift barrier model is that splicing error rate should vary not only across species according to *Ne*, but also among genes, according to their expression level. Indeed, for a given splicing error rate, the waste of resources (and hence the fitness cost) is expected to increase with the level of transcription. Thus, the selective pressure for optimal splice signals is expected to be higher, and hence the error rate to be lower, in highly expressed genes. Consistent with that prediction, we observed a negative correlation between gene expression level and AS rate in low-AS major-isoform introns in all but one species (*Figure 6C*).

It should be noted that our analyses suffer from several important limitations. First, the proxies that we considered for *Ne* are quite noisy (*Figure 1*). Second, to maximize the number of species in our analyses, we had to use very heterogeneous sources of RNA (whole-body, specific tissues, or organs, at different life stages, in different sexes, different environmental conditions, etc.). Third, we used short-read sequencing data, which allow the quantification of AS rates for individual introns, but do not provide a direct measure of AS rates *per* gene. Hopefully progress of long-read sequencing technologies will soon allow the comparative analysis of AS rates on full-length transcripts (e.g. see *Leung et al., 2021*). But presently, publicly available long-read transcriptomic data are restricted to a narrow set of model organisms, and their sequencing depth is still too limited to quantify rare splicing events. The fact that we detected significant correlations between AS rate and the three *Ne* proxies, despite these uncontrolled sources of variability, suggests that we underestimate the effect of *Ne* on AS rates.

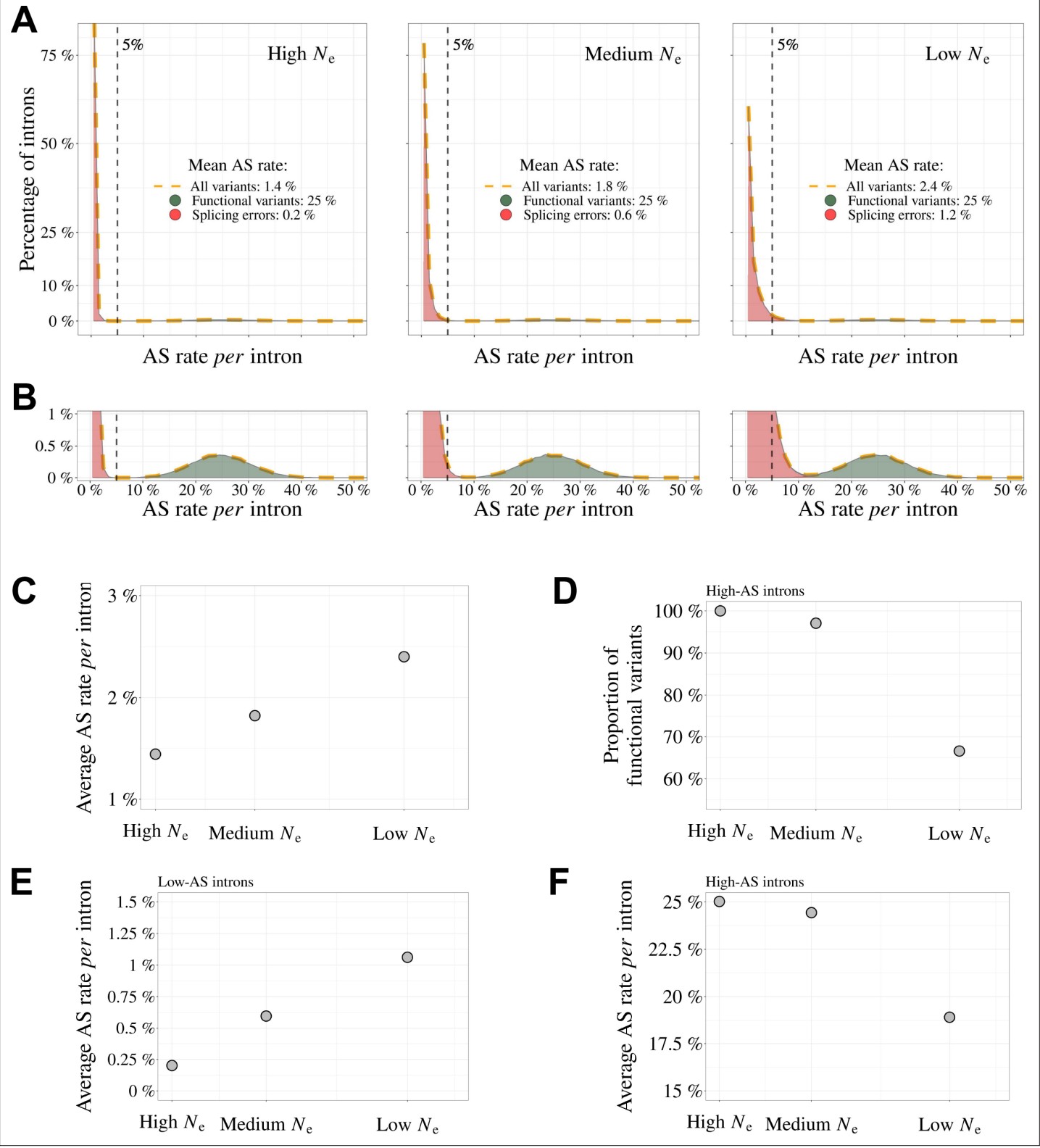

**Figure 7.** Impact of the drift-barrier on the genome-wide AS rate: model predictions. To illustrate the impact of the drift barrier, we sketched a simple model, with three hypothetical species of different *Ne*. In this model, the repertoire of SVs consists of a mixture of functional variants and splicing errors. We assumed that in all species, only a small fraction of major-isoform introns (5%) produce functional SVs, but that these variants have a relatively high AS rate (average = 25%, standard deviation = 5%; see Materials and methods for details on model settings). Splicing error rates were assumed to be

*Figure 7 continued on next page*

*Figure 7 continued*

gamma-distributed, with a low mean value. Owing to the drift barrier effect, the mean error rate was set to vary from 0.2% in species of high *Ne* to 1.2% in species of low *Ne* (these parameters were chosen to match approximately the AS rates observed in empirical data for rare SVs). (**A**) Genome-wide distribution of AS rates in each species (high *Ne*, medium *Ne* and low *Ne*). Each distribution corresponds to a mixture of functional SVs (green) and splicing errors (red). (**B**) Zoom on the y-axis to better visualize the contribution of functional SVs to the whole distribution: rare SVs (AS ≤ 5%) essentially correspond to splicing errors, while abundant SVs (AS > 5%) correspond to a mixture of functional and spurious variants, whose relative proportion depend on *Ne*. The following panels show how these different distributions, induced by differences in *Ne*, impact genome-wide AS patterns. (**C**) Relationship between the average AS rate *per* major-isoform intron and *Ne*. (**D**) Fraction of frame-preserving splice variants among introns with high AS rates *vs Ne*. Relationship between the average AS rate *per* intron and *Ne*, for 'low-AS' major-isoform introns (MIRA ≤ %) (**E**), and for 'high-AS' major-isoform introns (MIRA > 5%) (**F**).

Thus, overall, all observations fit qualitatively well with the predictions of the drift barrier model, according to which most of the variation in AS rate across species reflects differences in splicing error rates. Of course, this model is not in contradiction with the fact, well established, that some AS events play an essential role in various processes. Different criteria can be used to distinguish functional SVs from spurious splicing events. Notably, AS events that are strongly tissue-specific or developmentally dynamic tend to be more conserved across species, which indicates that a substantial fraction of them are evolutionary constrained, and hence functional (*Mudge et al., 2011*; *Barbosa-Morais et al., 2012*; *Merkin et al., 2012*; *Reyes et al., 2013*). The abundance of an SV is also an important predictor of its functionality. In particular, we observed that in all species, the proportion of frame-preserving events is much higher among abundant SVs than among rare SVs (*Figure 4A*). We note however that the threshold that we used to define abundant SVs is somewhat arbitrary. In fact, according to our model, this class of SVs corresponds to a mixture of functional and spurious events, whose relative proportion is expected to depend on *Ne* (*Figure 7C*). Thus, in low-*Ne* species, even the subset of abundant SVs includes a substantial fraction of errors. This probably explains why, contrarily to *Drosophila*, we do not detect any signature of purifying selection on alternative splice signals in humans, even for abundant SVs (*Figure 5*).

In conclusion, all observations fit with the hypothesis that random genetic drift sets an upper limit on the capacity of selection to prevent splicing errors. It should be noted that this limit on the optimization of genetic systems is expected to affect not only splicing, but all aspects of gene expression. Notably, there is a growing body of evidence that the complexity of transcripts produced by eukaryotic genes (resulting from alternative transcription initiation, polyadenylation, splicing or back-splicing, RNA editing) often does not correspond to fine-tuned adaptations but simply to the accumulation of errors (*Pickrell et al., 2010*; *Saudemont et al., 2017*; *Xu et al., 2019*; *Xu and Zhang, 2018*; *Liu and Zhang, 2018b*; *Liu and Zhang, 2018a*; *Xu and Zhang, 2014*; *Xu and Zhang, 2020*; *Gout et al., 2013*; *Zhang and Xu, 2022*). It should be noted however that the relationship between the genome-wide error rate and *Ne* is not expected to be monotonic. Indeed, models predict that in species with very high *Ne*, selection on each individual gene should favor genotypes that are robust to errors of the gene expression machinery, which in turn, reduces the constraints on the global level of gene expression errors (*Rajon and Masel, 2011*; *Xiong et al., 2017*). Thus, paradoxically, species with very large *Ne* are expected to have gene expression machineries that are more error-prone than species with very small *Ne* (*Rajon and Masel, 2011*). This argument was developed by *Xiong et al., 2017* to account for the fact that transcription error rates had been found to be about 10 times higher in bacteria than in eukaryotes (*Traverse and Ochman, 2016*; *Gout et al., 2013*). More recent work indicates that bacterial transcription error rates had been largely overestimated, presumably owing to RNA damages during the preparation of sequencing libraries (*Li and Lynch, 2020*). Given these uncertainties in the measures of transcription error rates, it seems for now difficult to interpret the differences reported across species. But in any case, it is important to note that it is in principle possible that the drift barrier affects differently the different steps of the gene expression process. It would therefore be important to investigate to which extent each step of gene expression responds (or not) to variation in *Ne*. As illustrated here by the relationship observed between alternative splicing and *Ne*, it appears essential to consider the contribution of non-adaptive evolutionary processes when trying to understand the origin of eukaryotic gene expression complexity.

## Materials and methods

### Genomic and transcriptomic data collection

To analyze AS rate variation across metazoans, three types of information are required: transcriptome sequencing (RNA-seq) datasets, genome assemblies, and gene annotations. To obtain this data, we first queried the Short Read Archive database (*Leinonen et al., 2011*) to extract publicly available RNA-seq datasets. We also queried the NCBI Genomes database (*Agarwala et al., 2018*) to retrieve genomic sequences and annotations. When this project was initiated, the vast majority of metazoans represented in this database corresponded to vertebrates or insects. We therefore decided to focus our analyses on these two clades (N=69 species).

### Identification of orthologous gene families

To be able to compare average AS rates across species, given that AS rates vary among genes (*Saudemont et al., 2017*), it is necessary to analyze a common set of orthologous genes. We searched for homologues of the BUSCOv3 (Benchmarking Universal Single Copy Orthologs *Seppey et al., 2019*) metazoan gene subset (N=978 genes) in each of the 69 genomes. To do this, we used the software BUSCO v.3.1.0 to associate BUSCO genes to annotated protein sequences. For each species, BUSCO genes were removed from the analysis if they were associated to more than one annotated gene or to an annotated gene that was associated to more than one BUSCO gene.

### RNA-seq data processing and intron identification

We aligned the RNA-seq reads on the corresponding reference genomes with HISAT2 v.2.1.0 (*Kim et al., 2019*). We built the genome indexes using annotated introns and exons coordinates in addition to genome sequences, to improve splice junction detection sensitivity. The maximum allowed intron length was fixed to 2,000,000 bp. We then extracted intron coordinates from HISAT2 alignments using an in-house perl script that scanned for CIGAR strings containing N, which indicate regions that are skipped from the reference sequence. For intron detection and quantification we used only uniquely mapping reads that had a maximum mismatch ratio of 0.02. We required a minimum anchor length (that is, the number of bases that align on each flanking exon) of 8 bp for intron detection, and of 5 bp for intron quantification. We kept only those predicted introns that had GT-AG, GC-AG or AT-AC splice signals, and we predicted the strand of the introns based on the splice signal.

We assigned an intron to a gene if at least one of the intron boundaries fell within 1 bp of the annotated exon coordinates of the gene, combined across all annotated isoforms. We excluded introns that could not be unambiguously assigned to a single gene. We distinguish annotated introns (which appear as such in the reference genome annotations) and un-annotated introns, which were detected with RNA-seq data and assigned to previously annotated genes.

We further restricted our analyses to introns located within protein-coding regions. To do this, for each protein-coding gene, we extracted the start codons and the stop codons for all annotated isoforms. We then identified the minimum start codon and the maximum end codon positions and we excluded introns that were upstream or downstream of these extreme coordinates.

The alignment process, which is the most time-consuming step in the pipeline (see *Figure 2—figure supplement 3*), can take up to 1 week when using 16 cores *per* RNA-seq for larger genomes, such as mammals. Additionally, the processed compressed files generated during this process can exceed 7 terabytes in size.

### Alternative splicing rate definition

For each intron we noted $N_s$ the number of reads corresponding to the precise excision of this intron (spliced reads), and $N_a$ the number of alternatively spliced reads (i.e. spliced variant sharing only one of the two intron boundaries). Finally, we note $N_u$ the number of unspliced reads, co-linear with the genomic sequence, and which overlap with at least 10 bp on each side of an exon-intron boundary. These definitions are illustrated in *Figure 2*. We then defined the relative abundance of the focal intron compared to introns with one alternative splice boundary (RAS $= \frac{N_s}{N_s + N_a}$), as well as relative to unspliced reads (RANS $= \frac{N_s}{N_s + \frac{N_u}{2}}$).

To compute these ratios we required a minimal number of 10 reads at the denominator. We thus calculated the RAS only if $(N_s + N_a) \geq 10$ and the RANS only if $(N_s + \frac{N_u}{2}) \geq 10$ (We divided $N_u$ by 2

because retention is quantified at two sites, which increases the detection power by a factor of 2). If the criteria were not met, the values were labeled as not available (NA). We computed these ratios using reads from all available RNA-seq samples, unless otherwise specified (e.g. in sub-sampling analyses). Based on these ratios, we defined three categories of introns: major-isoform introns, defined as those introns that have RANS > 0.5 and RAS > 0.5; minor-isoform introns, defined as those introns that have RANS ≤ 0.5 or RAS ≤ 0.5; unclassified introns, which do not satisfy the above conditions.

We determined the alternative splicing (AS) rate of major-isoform introns using the following formula: $AS = \frac{N^M}{N^M + N^m}$, where $N^M$ is the number of spliced reads corresponding to the excision of the major-isoform intron and $N^m$ is the total number of spliced reads corresponding to the excision of minor-isoform introns sharing a boundary with a major-isoform intron (see *Figure 2*).

For minor-isoform introns sharing a boundary with a major-isoform intron, we computed the relative abundance of the minor-isoform intron (i) with respect to the corresponding major-isoform intron, with the following formula: $\text{Minor intron relative abundance } MIRA_i = \frac{N_i^m}{N^M + N^m}$, where $N_i^m$ is the number of spliced reads corresponding to the excision of a minor-isoform intron (i) (see *Figure 2*).

We defined the *per*-gene AS rate as the probability to observe at least one alternative splicing event across all the major-isoform introns of a gene. To estimate the per-gene AS rate of a given gene, we assumed that the AS rate is uniform across its major-isoform introns, and that AS events occur independently at each intron. We calculated the AS rate for each gene as the number of spliced reads corresponding to the excision of major-isoform introns, divided by the number of spliced reads corresponding to minor and major-isoform introns ($\frac{\sum N^m}{\sum N^M + N^m}$). The probability for a given gene to produce no splice variant across all its major-isoform introns is thus $p0 = (1 - \frac{\sum N^m}{\sum N^M + N^m})^{N_i}$, where $N_i$ is the number of major-isoform introns of the gene. The *per*-gene AS rate (ASg), that is the probability to have at least one AS event, is therefore the complement of p0: ASg = 1-p0.

## Identification of reading frame-preserving splice variants

To determine the proportion of open-reading frame-preserving splice variants, we first identified minor-isoform introns that had their minor splice site within a maximum distance of 30 bp from the major splice site (either in the flanking exon or within the major-isoform intron). We chose this length threshold because it is shorter than the size of the smallest introns in metazoans, so that to avoid the possibility of having a skipped exon between the minor and the major splice site (which could induce some ambiguities in the assessment of the reading frame). Among these introns, we considered that frame-preserving variants are those introns for which the distance between the minor-isoform intron boundary and the major-isoform intron boundary was a multiple of 3.

## Gene expression level

Gene expression levels were calculated with Cufflinks v2.2.1 (*Roberts et al., 2011*) based on the read alignments obtained with HISAT2, for each RNA-seq sample individually. We estimated FPKM levels (Fragments *Per* Kilobase of exon *per* Million mapped reads) for each gene.

The overall gene expression of a gene was computed as the average FPKM across samples, weighted by the sequencing depth of each sample. The sequencing depth of a sample is the median *per*-base read coverage across BUSCO genes.

## Phylogenetic tree reconstruction

For each of the 978 BUSCO gene families we collected the longest corresponding proteins identified in each species. We removed proteins for which the amino acid sequence provided with the annotations did not perfectly correspond to the translation of the corresponding coding sequences. We then aligned the resulting sets of protein-coding sequences for each BUSCO gene, using the codon alignment option in PRANK v.170427 (*Löytynoja and Goldman, 2008*). We translated the codon alignments into protein alignments using the R package seqinr (*Charif and Lobry, 2007*).

To infer the phylogenetic tree rapidly, we sub-sampled the resulting multiple alignments (N=461), selecting alignments with the highest number of species (ranging from 49 to 53 species *per* alignment). We then concatenated these alignments and kept sites that were aligned in at least 30 species. We used RAxML-NG v.0.9.0 (*Kozlov et al., 2019*) to infer the species phylogeny with a final alignment of 53 taxa and 165,648 sites (amino acids). RAxML was set to perform one model *per* gene with fixed empirical substitution matrix (LG), empirical amino acid frequencies from alignment (F) and 8 discrete

GAMMA categories (G8), specified in a partition file with one line *per* multiple alignment. The analysis generated 10 starting trees, 5 starting from a random topology and 5 starting from a tree generated by the parsimony-based randomized stepwise addition algorithm. The best-scoring topology was kept as the final ML tree and 10 bootstrap replicates have been generated.

## *dN/dS* computation

We estimated *dN/dS* ratios for the BUSCO gene families that were present in at least 45 species (N=922 genes), using the codon alignments obtained with PRANK (see above). We divided the 922 sequence alignments into 18 groups, based on their average GC3 content across species, and concatenated the alignments within each group. We thus obtained concatenated alignments that were 209 kb long on average. We used bio ++v.3.0.0 libraries (*Guéguen et al., 2013*; *Dutheil and Boussau, 2008*; *Bolívar et al., 2019*) to estimate the *dN/dS* on terminal branches of the phylogenetic tree, for each concatenated alignment. We attributed the *dN/dS* of the terminal branches to the species that corresponds.

In a first step, we used an homogeneous codon model implemented in bppml to infer the most likely branch lengths, codon frequencies at the root, and substitution model parameters. We used YN98 (F3X4) (*Yang and Nielsen, 1998*) substitution model, which allows for different nucleotide content dynamics across codon positions. In a second step, we used the MapNH substitution mapping method (*Guéguen and Duret, 2018*) to count synonymous and non-synonymous substitutions (*Dutheil et al., 2012*). We defined dN as the total number of non-synonymous substitutions divided by the total number of non-synonymous opportunities, both summed across concatenated alignments, for each branch of the phylogenetic tree. Likewise, we defined dS as the total number of synonymous substitutions divided by the total number of synonymous opportunities, both summed across concatenated alignments. The *per*-species *dN/dS* corresponds to the ratio between dN and dS, on the terminal branches of the phylogenetic tree.

## Life history traits

We used various life history traits to approximate the effective population size of each species. For vertebrates species we considered the maximum lifespan (i.e. from birth to death) and body length referenced. For insects we took the maximum lifespan and body length of the *imago*. For eusocial insects and the eusocial mammal *Heterocephalus glaber*, the selected values correspond to the queens. The sources from which the lifespan and the body length information was taken are listed in data/Data9-supp.pdf in the Zenodo repository (see Data and code availability).

## Analyses of sequence polymorphism

We analyzed the distribution of single-nucleotide polymorphisms (SNPs) around splice sites in *Drosophila melanogaster* and *Homo sapiens*.

For *D. melanogaster*, we used polymorphism data from the *Drosophila* Genetic Reference Panel (DGRP; *Mackay et al., 2012*; *Huang et al., 2014*), from which we extracted 3,963,397 SNPs that were identified from comparisons across 205 inbred lines. We converted the SNP coordinates from the dm3 genome assembly to the dm6 assembly with the liftOver utility (*Hinrichs et al., 2006*) of the UCSC genome browser, using a whole genome alignment between the two assemblies downloaded here.

For *H. sapiens*, we used polymorphism data from the 1000 Genomes project, phase 3 release (*Auton et al., 2015*). This dataset included 80,868,061 SNPs that were genotyped in 2,504 individuals.

For each minor-isoform intron sharing one boundary with a major-isoform intron, we computed the number of SNPs that occur at their respective splice sites: at their shared boundary, and at the major-isoform intron and minor-isoform introns specific boundaries.

We focused our study on minor-isoform introns that have their specific boundary folding in the exons adjacent to the major-isoform intron or in the major-isoform intron. As a control, for each minor-isoform intron, we searched for one GT and one AG dinucleotides in the interval between 20 and 60 bp with respect to the major splice site, in the neighboring exon and in the major-isoform intron, and computed the number of SNPs that occur on these sites. We searched for control AG dinucleotides in the vicinity of the donor splice site of the major-isoform intron and for GT dinucleotides in the vicinity of its acceptor splice site, to avoid studying sites that might correspond to unidentified minor splice sites. For *H. sapiens*, we further divided the splice sites and the control dinucleotides into two groups, depending on whether they were subject to CpG hypermutability or not.

## Impact of the drift-barrier on genome-wide AS rates: sketched model

To illustrate the impact of the drift barrier, we sketched a simple model, with three hypothetical species of different $N_e$ (low, medium, and high $N_e$). In each species, the repertoire of SVs consists of two categories: functional variants and spurious variants (which result from errors of the splicing machinery). The rate of splicing error was assumed to be low and to depend on $N_e$, owing to the drift barrier effect. We considered that in all species, only a small fraction of major-isoform introns (5%) produce functional SVs, but that these variants have a relatively high AS rate. The AS rates of functional SVs were modeled by a normal distribution, with a mean of 25% and a standard deviation of 5% (same parameters for the three species). We modeled the distribution of error rates by a gamma distribution, with shape parameter = 1, and with mean values of 0.2%, 0.6% and 1.2% respectively in species of high, medium or low $N_e$ (these parameters were set to match approximately the AS rates observed in empirical data for rare SVs). We then combined the two distributions (functional SVs and splicing errors) to compute the genome-wide average AS rates in each species. We also computed the average AS rate on the subsets of low-AS or high-AS major-isoform introns (i.e. with AS rates respectively below or above the threshold AS rate of 5%). Finally, we computed the proportion of frame-preserving SVs among high-AS major-isoform introns, assuming that two thirds of splicing errors induce frameshifts and that all functional SVs preserve the reading frame.

## Data and code availability

All processed data that we generated and used in this study, as well as the scripts that we used to analyze the data and to generate the figures, are available on Zenodo DOI: https://doi.org/10.5281/zenodo.7415114.

In particular, the sources of transcriptomic data, genome assemblies and annotations are reported in the Zenodo archive in data/Data1-supp.tab. The archive includes several directories, including figure, which contains the necessary materials to produce the figures of the manuscript. Rmarkdown scripts located in the table_supp directory were used to generate supplementary tables, which are also saved in the same directory. The processed data used to generate figures and conduct analyses are stored in the data directory in tab-separated text format.

## Acknowledgements

We thank Loïc Guille for his contribution to an initial pilot study, Tristan Lefébure for insightful discussions and Laurent Guéguen for his help on dN/dS analyses. Computational analyses were performed using the computing facilities of the CC LBBE/PRABI and the Core Cluster of the Institut Français de Bioinformatique (IFB) (ANR-11-INBS-0013). We thank five anonymous reviewers for their thorough and constructive comments, which were very helpful to improve our manuscript. A preprint version of this article has been peer-reviewed and recommended by PCIEvolBiol (https://doi.org/10.24072/pci.evolbiol.100642). This work was funded by the French National Research Agency (ANR-20-CE02-0008-01 "NeGA" and ANR-17-CE12-0019-01 "LncEvoSys").

## Additional information

### Funding

| Funder | Grant reference number | Author |
| --- | --- | --- |
| Agence Nationale de la Recherche | ANR-20-CE02-0008-01 | Florian Bénitière<br>Anamaria Necsulea<br>Laurent Duret |
| Agence Nationale de la Recherche | ANR-17-CE12-0019-01 | Anamaria Necsulea<br>Laurent Duret |

The funders had no role in study design, data collection and interpretation, or the decision to submit the work for publication.

## Author contributions

Florian Bénitière, Resources, Data curation, Software, Formal analysis, Investigation, Methodology, Writing - original draft, Writing - review and editing; Anamaria Necsulea, Conceptualization, Resources, Data curation, Software, Formal analysis, Supervision, Funding acquisition, Investigation, Methodology, Writing - original draft, Project administration, Writing - review and editing; Laurent Duret, Conceptualization, Formal analysis, Supervision, Funding acquisition, Investigation, Methodology, Writing - original draft, Project administration, Writing - review and editing

## Author ORCIDs

Florian Bénitière http://orcid.org/0000-0001-7773-3542
Anamaria Necsulea http://orcid.org/0000-0001-9861-7698
Laurent Duret http://orcid.org/0000-0003-2836-3463

Reviewer #1 (Public Review): https://doi.org/10.7554/eLife.93629.3.sa1
Reviewer #2 (Public Review): https://doi.org/10.7554/eLife.93629.3.sa2
Author Response https://doi.org/10.7554/eLife.93629.3.sa3

## Additional files

### Supplementary files
• MDAR checklist

### Data availability

All processed data that we generated and used in this study, as well as the scripts that we used to analyze the data and to generate the figures, are available on zenodo DOI: https://doi.org/10.5281/zenodo.7415114.

The following dataset was generated:

| Author(s) | Year | Dataset title | Dataset URL | Database and Identifier |
|---|---|---|---|---|
| Bénitière F, Necsulea A, Duret L | 2024 | 2023-Random genetic drift sets an upper limit on mRNA splicing accuracy in metazoans | https://zenodo.org/doi/10.5281/zenodo.7415114 | Zenodo, 10.5281/zenodo.7415114 |

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
