## [Editor Report · eLife assessment]

This **fundamental** study evaluates the evolutionary significance of variations in the accuracy of the intron-splicing process across vertebrates and insects. Using a powerful combination of comparative and population genomics approaches, the authors present **convincing** evidence that higher rates of alternative splicing tend to be observed in species with lower effective population size, a key prediction of the drift-barrier hypothesis. The analysis is carefully conducted and has broad implications beyond the studied species. As such, it will strongly appeal to anyone interested in the evolution of genome architecture and the optimisation of genetic systems.

---

## [Referee Report · Reviewer #1 (Public Review)]

Summary:

Functionally important alternative isoforms are gold nuggets found in a swamp of errors produced by the splicing machinery.

The architecture of eukaryotic genomes, when compared with prokaryotes, is characterised by a preponderance of introns. These elements, which are still present within transcripts, are rapidly removed during the splicing of messenger RNA (mRNA), thus not contributing to the final protein. The extreme rarity of introns in prokaryotes, and the elimination of these introns from mRNAs before translation into protein, raises questions about the function of introns in genomes. One explanation comes from functional biology: introns are thought to be involved in post-transcriptional regulation and in the production of translational variants. The latter function is possible when the positions of the edges of the spliced intron vary. While some light has been shed on specific examples of the functional role of alternative splicing, to what extent are they representative of all introns in metazoans?

In this study, the hypothesis of a functional role for alternative splicing, and therefore to a certain extent for introns, is evaluated against another explanation coming from evolutionary biology: isoforms are above all errors of imprecision by the molecular machinery at work during splicing. This hypothesis is based on a principle established by Motoo Mikura, which has become central to population genetics, explaining that the evolutionary trajectory of a mutation with a given effect is intimately linked to the effective population size (Ne) where this mutation emerges. Thus, the probability of fixation of a weakly deleterious mutation increases when Ne decreases, and the probability of fixation of a weakly advantageous mutation increases when Ne increases. The genomes of populations with low Ne are therefore expected to accumulate more weakly deleterious mutations and fewer weakly advantageous mutations than populations with high Ne. In this framework, if splicing errors have only small effects on the fitness of individuals, then natural selection cannot increase the precision of the splicing machinery, allowing tolerance for the production of alternative isoforms.

In the past, the debate opposed one-off observations of effectively functional isoforms on the one hand, to global genomic quantities describing patterns without the possibility of interpreting them in detail. The authors here propose an elegant quantitative approach in line with the expected continuous variation in the effectiveness of selection, both between species and within genomes. The result describing the inter-specific pattern on a large scale confirms what was already known (there is a negative relationship between effective size and average alternative splicing rate). The essential novelty of this study lies in (1) the quantification, for each intron studied, of the relative abundance of each isoform, and (2) the analysis of a relationship between this abundance and the evolutionary constraints acting on these isoforms.

What is striking is the light shed on the general very low abundance of alternative isoforms. Depending on the species, 60% to 96% of cases of alternatively spliced introns lead to an isoform whose abundance is less than 5% of the total variants for a given intron.

In addition to the fact that 60%-96% of the total isoforms are more than 20 times less abundant than their majority form, this large proportion of alternative isoforms exhibit coding-phase shift at rates similar to what would be expected by chance, i.e. for a third of them, which reinforces the idea that there is no particular constraint on these isoforms.

The remaining 4%-40% of isoforms see their coding-phase shift rate decrease as their relative abundance increases. This result represents a major step forward in our understanding of alternative splicing and makes it possible to establish a quantitative model directly linking the relative abundance of an isoform with a putative functional role concerning only those isoforms produced in abundance. Only the (rare) isoforms which are abundantly produced are thought to be involved in a biological function.

Within the same genome, the authors show that only highly expressed genes, i.e. those that tend to be more constrained on average, are also the genes with the lowest alternative splicing rates on average.

The comparison between species in this study reveals that the smaller the effective size of a species, the more its genome produces isoforms that are low in abundance and low in constraint. Conversely, species with a large effective size relatively reduce rare isoforms, and increase stress on abundant isoforms.

To sum up:

• the higher the effective size of a species, the fewer introns are spliced.

• highly expressed genes are spliced less.

• when splicing occurs, it is mainly to produce low-abundance isoforms.

• low-abundance isoforms are also less constrained.

Taken together, these results reinforce a quantitative view of the evolution of alternative splicing as being mainly the product of imprecision in the splicing machinery, generating a great deal of molecular noise. Then, out of all this noise, a few functional gold nuggets can sometimes emerge. From the point of view of the reviewer, the evolutionary dynamics of genomes are depressing. The small effective population sizes are responsible for the accumulation of multiple slightly deleterious introns. Admittedly, metazoan genomes try to get rid of these introns during RNA maturation, but this mechanism is itself rendered imprecise by population sizes.

Strengths:

• The authors simultaneously study the effects of effective population size, isoform abundance, and gene expression levels on the evolutionary constraints acting on isoforms. Within this framework, they clearly show that an isoform becomes functionally important only under certain rare conditions.

• The authors rule out an effect putatively linked to variations in expression between different organs which could have biased comparisons between different species.

Weaknesses:

• While the longevity of organisms as a measure of effective size seems to work overall, it may not be relevant for discriminating within a clade. For example, within Hymenoptera, we might expect them to have the same overall longevity, but that effective size would be influenced more by the degree of sociality: solitary bees/ants/wasps versus eusocial. I am therefore certain that the relationship shown in Figure 4D is currently not significant because the measure of effective size is not relevant for Hymenoptera. The article would have been even more convincing by contrasting the rates of alternative splicing between solitary versus social hymenopterans.

• When functionalist biologists emphasise the role of the complexity of living things, I'm not sure they're thinking of the comparison between "*Drosophila*" and "*Homo sapiens*", but rather of a broader evolutionary scale. Which gives the impression of an exaggeration of the debate in the introduction.

---

## [Referee Report · Reviewer #2 (Public Review)]

Summary:

Two hypotheses could explain the observation that genes of more complex organisms tend to undergo more alternative splicing. On one hand, alternative splicing could be adaptive since it provides the functional diversity required for complexity. On the other hand, increased rates of alternative splicing could result through nonadaptive processes since more complex organisms tend to have smaller effective population sizes and are thus more prone to deleterious mutations resulting in more spurious splicing events (drift-barrier hypothesis). To evaluate the latter, B́enitiere et al. analyzed transcriptome sequencing data across 53 metazoan species. They show that proxies for effective population size and alternative splicing rates are negatively correlated. Furthermore, the authors find that rare, nonfunctional (and likely erroneous) isoforms occur more frequently in more complex species. Additionally, they show evidence that the strength of selection on splice sites increases with increasing effective population size and that the abundance of rare splice variants decreases with increased gene expression. All of these findings are consistent with the drift-barrier hypothesis.

This study conducts a comprehensive set of separate analyses that all converge on the same overall result and the manuscript is well organized. Furthermore, this study is useful in that it provides a modified null hypothesis that can be used for future tests of adaptive explanations for variation in alternative splicing.

Strengths:

The major strength of this study lies in its complementary approach combining comparative and population genomics. Comparing evolutionary trends across phylogenetic diversity is a powerful way to test hypotheses about the origins of genome complexity. This approach alone reveals several convincing lines of evidence in support of the drift-barrier hypothesis. However, the authors also provide evidence from a population genetics perspective (using resequencing data for humans and fruit flies), making results even more convincing.

The authors are forward about the study's limitations and explain them in detail. They elaborate on possible confounding factors as well as the issues with data quality (e.g. proxies for Ne, inadequacies of short reads, heterogeneity in RNA-sequencing data).

Weaknesses:

The authors primarily consider insects and mammals in their study. This only represents a small fraction of metazoan diversity. Sampling from a greater diversity of metazoan lineages would make these results and their relevance to broader metazoans substantially more convincing. Although the authors are careful about their tone, it is challenging to reconcile these results with trends across greater metazoans when the underlying dataset exhibits ascertainment bias and represents samples from only a few phylogenetic groups. Relatedly, some trends (such as Figure 1B-C) seem to be driven primarily by non-insect species, raising the question of whether some results may be primarily explained by specific phylogenetic groups (although the authors do correct for phylogeny in their statistics). How might results look if insects and mammals (or vertebrates) are considered independently?

Throughout the manuscript, the authors refer to infrequently spliced (mode <5%) introns as "minor introns" and frequently spliced (mode >95%) as "major introns". This is extremely confusing since "minor introns" typically represent introns spliced by the U12 spliceosome, whereas "major introns" are those spliced by the U2 spliceosome. Furthermore, it remains unclear whether the study only considers major introns or both major and minor introns. Minor introns typically have AT-AC splice sites whereas major introns usually have GT/GC-AG splice sites, although in rare cases the U2 can recognize AT-AC (see Wu and Krainer 1997 for example). The authors also note that some introns show noncanonical AT-AC splice sites while these are actually canonical splice sites for minor introns.

---

## [Author Response]

The following is the authors’ response to the original reviews.

eLife assessmentThis fundamental study evaluates the evolutionary significance of variations in the accuracy of the intron-splicing process across vertebrates and insects. Using a powerful combination of comparative and population genomics approaches, the authors present convincing evidence that species with lower effective population size tend to exhibit higher rates of alternative splicing, a key prediction of the drift-barrier hypothesis. The analysis is carefully conducted and all observations fit with this hypothesis, but focusing on a greater diversity of metazoan lineages would make these results even more broadly relevant. This study will strongly appeal to anyone interested in the evolution of genome architecture and the optimisation of genetic systems.
**Public Reviews:**

**Reviewer #1 Public Review:**
Summary:Functionally important alternative isoforms are gold nuggets found in a swamp of errors produced by the splicing machinery.The architecture of eukaryotic genomes, when compared with prokaryotes, is characterised by a preponderance of introns. These elements, which are still present within transcripts, are rapidly removed during the splicing of messenger RNA (mRNA), thus not contributing to the final protein. The extreme rarity of introns in prokaryotes, and the elimination of these introns from mRNAs before translation into protein, raises questions about the function of introns in genomes. One explanation comes from functional biology: introns are thought to be involved in post-transcriptional regulation and in the production of translational variants. The latter function is possible when the positions of the edges of the spliced intron vary. While some light has been shed on specific examples of the functional role of alternative splicing, to what extent are they representative of all introns in metazoans?In this study, the hypothesis of a functional role for alternative splicing, and therefore to a certain extent for introns, is evaluated against another explanation coming from evolutionary biology: isoforms are above all errors of imprecision by the molecular machinery at work during splicing. This hypothesis is based on a principle established by Motoo Kimura, which has become central to population genetics, explaining that the evolutionary trajectory of a mutation with a given effect is intimately linked to the effective population size (Ne) where this mutation emerges. Thus, the probability of fixation of a weakly deleterious mutation increases when Ne decreases, and the probability of fixation of a weakly advantageous mutation increases when Ne increases. The genomes of populations with low Ne are therefore expected to accumulate more weakly deleterious mutations and fewer weakly advantageous mutations than populations with high Ne. In this framework, if splicing errors have only small effects on the fitness of individuals, then natural selection cannot increase the precision of the splicing machinery, allowing tolerance for the production of alternative isoforms.In the past, the debate opposed one-off observations of effectively functional isoforms on the one hand, to global genomic quantities describing patterns without the possibility of interpreting them in detail. The authors here propose an elegant quantitative approach in line with the expected continuous variation in the effectiveness of selection, both between species and within genomes. The result describing the inter-specific pattern on a large scale confirms what was already known (there is a negative relationship between effective size and average alternative splicing rate). The essential novelty of this study lies in (1) the quantification, for each intron studied, of the relative abundance of each isoform, and (2) the analysis of a relationship between this abundance and the evolutionary constraints acting on these isoforms.What is striking is the light shed on the general very low abundance of alternative isoforms. Depending on the species, 60% to 96% of cases of alternatively spliced introns lead to an isoform whose abundance is less than 5% of the total variants for a given intron.In addition to the fact that 60 %-96% of the total isoforms are more than 20 times less abundant than their majority form, this large proportion of alternative isoforms exhibit coding-phase shift at rates similar to what would be expected by chance, i.e. for a third of them, which reinforces the idea that there is no particular constraint on these isoforms.The remaining 4%-40% of isoforms see their coding-phase shift rate decrease as their relative abundance increases. This result represents a major step forward in our understanding of alternative splicing and makes it possible to establish a quantitative model directly linking the relative abundance of an isoform with a putative functional role concerning only those isoforms produced in abundance. Only the (rare) isoforms which are abundantly produced are thought to be involved in a biological function.Within the same genome, the authors show that only highly expressed genes, i.e. those that tend to be more constrained on average, are also the genes with the lowest alternative splicing rates on average.The comparison between species in this study reveals that the smaller the effective size of a species, the more its genome produces isoforms that are low in abundance and low in constraint. Conversely, species with a large effective size relatively reduce rare isoforms, and increase stress on abundant isoforms.To sum up:the higher the effective size of a species, the fewer introns are spliced.highly expressed genes are spliced less.when splicing occurs, it is mainly to produce low-abundance isoforms.low-abundance isoforms are also less constrained.Taken together, these results reinforce a quantitative view of the evolution of alternative splicing as being mainly the product of imprecision in the splicing machinery, generating a great deal of molecular noise. Then, out of all this noise, a few functional gold nuggets can sometimes emerge. From the point of view of the reviewer, the evolutionary dynamics of genomes are depressing. The small effective population sizes are responsible for the accumulation of multiple slightly deleterious introns. Admittedly, metazoan genomes try to get rid of these introns during RNA maturation, but this mechanism is itself rendered imprecise by population sizes.Strengths:The authors simultaneously study the effects of effective population size, isoform abundance, and gene expression levels on the evolutionary constraints acting on isoforms. Within this framework, they clearly show that an isoform becomes functionally important only under certain rare conditions.The authors rule out an effect putatively linked to variations in expression between different organs which could have biased comparisons between different species.Weaknesses:While the longevity of organisms as a measure of effective size seems to work overall, it may not be relevant for discriminating within a clade. For example, within Hymenoptera, we might expect them to have the same overall longevity, but that effective size would be influenced more by the degree of sociality: solitary bees/ants/wasps versus eusocial. I am therefore certain that the relationship shown in Figure 4D is currently not significant because the measure of effective size is not relevant for Hymenoptera. The article would have been even more convincing by contrasting the rates of alternative splicing between solitary versus social hymenopterans.

As suggested by the reviewer, we investigated the degree of sociality for the 18 hymenopterans included in our study. We observed that the average dN/dS of the 12 eusocial species (4 bees, 6 ants, 2 wasps) is significantly higher than that of the 6 solitary species (p=2.1x10-3; Fig. R1A), consistent with a lower effective population size in eusocial species compared to solitary ones.

However, the AS rate does not differ significantly between these two groups, neither for the full set of major-isoform introns (Fig. R1B), nor for the subsets of low-AS or high-AS major-isoform introns (Fig. R1C,D). Given the limited sample size (12 eusocial species, 6 solitary species), it is possible that some uncontrolled variables affecting the AS rate hide the impact of Ne.

**Author response image 1. sa3fig1:** Comparison of solitary (N=6) and eusocial hymenopterans (N=12). (**A**) dN/dS ratio.(**B**) AS rate (all major-isoform introns). (**C**) AS rate (low-AS major-isoform introns). (**D**) AS rate (high-AS major-isoform introns). The means of the two group were compared with a Wilcoxon test.

When functionalist biologists emphasise the role of the complexity of living things, I'm not sure they're thinking of the comparison between "*Drosophila*" and "*Homo sapiens*", but rather of a broader evolutionary scale. Which gives the impression of an exaggeration of the debate in the introduction.

We disagree with the referee: in fact, all the debate regarding the paradox of the absence of relationship between the number of genes and organismal complexity arose from the comparative analysis of gene repertoires across metazoans. This debate started in the early 2000’s, when the sequencing of the human genome revealed that it contains only ~20,000 protein-coding genes (far less than the ~100,000 genes that were expected at that time). This came as a big surprise because it showed that the gene repertoire of mammals is not larger than that of invertebrates such as Caenorhabditis elegans (19,000 genes) or *Drosophila melanogaster* (14,000 genes) . We cite below several articles that illustrate how this paradox has been perceived by the scientific community:

Graveley BR 2001 Alternative splicing: increasing diversity in the proteomic world. Trends in Genetics 17 : 100–107. https://doi.org/10.1016/S0168-9525(00)02176-4

“ How can the genome of *Drosophila melanogaster* contain fewer genes than the undoubtedly simpler organism Caenorhabditis elegans? ”

Ewing B and Green P 2000 Analysis of expressed sequence tags indicates 35,000 human genes. Nature Genetics 25 : 232–234. https://doi.org/10.1038/76115

“ the invertebrates Caenorhabditis elegans and *Drosophila melanogaster* having 19,000 and 13,600 genes, respectively. Here we estimate the number of human genes […] approximately 35,000 genes, substantially lower than most previous estimates. Evolution of the increased physiological complexity of vertebrates may therefore have depended more on the combinatorial diversification of regulatory networks or alternative splicing than on a substantial increase in gene number. ”

Kim E, Magen A and Ast G 2007 Different levels of alternative splicing among eukaryotes. Nucleic Acids Research 35 : 125–131. https://doi.org/10.1093/nar/gkl924

“we reveal that the percentage of genes and exons undergoing alternative splicing is higher in vertebrates compared with invertebrates. […] The difference in the level of alternative splicing suggests that alternative splicing may contribute greatly to the mammal higher level of phenotypic complexity,”

Nilsen TW and Graveley BR 2010 Expansion of the eukaryotic proteome by alternative splicing. Nature 463 : 457–463. https://doi.org/10.1038/nature08909

“ It is noteworthy that *Caenorhabditis elegans*, *D. melanogaster* and mammals have about 20,000 (ref. 68), 14,000 (ref. 69) and 20,000 (ref. 70) genes, respectively, but mammals are clearly much more complex than nematodes or flies.”

**Reviewer #2 (Public Review):**
Summary:Two hypotheses could explain the observation that genes of more complex organisms tend to undergo more alternative splicing. On one hand, alternative splicing could be adaptive since it provides the functional diversity required for complexity. On the other hand, increased rates of alternative splicing could result through nonadaptive processes since more complex organisms tend to have smaller effective population sizes and are thus more prone to deleterious mutations resulting in more spurious splicing events (drift-barrier hypothesis). To evaluate the latter, Bénitière et al. analyzed transcriptome sequencing data across 53 metazoan species. They show that proxies for effective population size and alternative splicing rates are negatively correlated. Furthermore, the authors find that rare, nonfunctional (and likely erroneous) isoforms occur more frequently in more complex species. Additionally, they show evidence that the strength of selection on splice sites increases with increasing effective population size and that the abundance of rare splice variants decreases with increased gene expression. All of these findings are consistent with the drift-barrier hypothesis.This study conducts a comprehensive set of separate analyses that all converge on the same overall result and the manuscript is well organized. Furthermore, this study is useful in that it provides a modified null hypothesis that can be used for future tests of adaptive explanations for variation in alternative splicing.Strengths:The major strength of this study lies in its complementary approach combining comparative and population genomics. Comparing evolutionary trends across phylogenetic diversity is a powerful way to test hypotheses about the origins of genome complexity. This approach alone reveals several convincing lines of evidence in support of the drift-barrier hypothesis. However, the authors also provide evidence from a population genetics perspective (using resequencing data for humans and fruit flies), making results even more convincing.The authors are forward about the study's limitations and explain them in detail. They elaborate on possible confounding factors as well as the issues with data quality (e.g. proxies for Ne, inadequacies of short reads, heterogeneity in RNA-sequencing data).Weaknesses:The authors primarily consider insects and mammals in their study. This only represents a small fraction of metazoan diversity. Sampling from a greater diversity of metazoan lineages would make these results and their relevance to broader metazoans substantially more convincing. Although the authors are careful about their tone, it is challenging to reconcile these results with trends across greater metazoans when the underlying dataset exhibits ascertainment bias and represents samples from only a few phylogenetic groups. Relatedly, some trends (such as Figure 1B-C) seem to be driven primarily by non-insect species, raising the question of whether some results may be primarily explained by specific phylogenetic groups ( although the authors do correct for phylogeny in their statistics). How might results look if insects and mammals (or vertebrates) are considered independently?

Following the referee’s suggestion, we investigated the relationship between AS rate and proxies of Ne, separately for insects and vertebrates (Supplementary Fig. 11) . We observed that the relationship was consistent in vertebrates and insects: linear regressions show a positive correlation, significant (p<0.05) in all cases, except for body length in vertebrates. We added a sentence (line 166) to mention this point.

Note that for these analyses we have smaller sample sizes, so we have a weaker power to detect signal. We therefore prefer to present the combined analyses, using PGLS to account for phylogenetic inertia.

Throughout the manuscript, the authors refer to infrequently spliced ( mode <5%) introns as "minor introns" and frequently spliced (mode >95%) as "major introns". This is extremely confusing since "minor introns" typically represent introns spliced by the U12 spliceosome, whereas "major introns" are those spliced by the U2 spliceosome.

To avoid any confusion, we modified the terminology: we now refer to infrequently spliced introns as " minor-isoform introns" and frequently spliced as "major -isoform introns" (see line 135-137) . The entire manuscript (including the figures) has been modified accordingly.

Furthermore, it remains unclear whether the study only considers major introns or both major and minor introns. Minor introns typically have AT-AC splice sites whereas major introns usually have GT/GC-AG splice sites, although in rare cases the U2 can recognize AT-AC (see Wu and Krainer 1997 for example).

We modified the text (line 148-150) to clearly state that we studied all introns, both U2-type and U12-type.

The authors also note that some introns show noncanonical AT-AC splice sites while these are actually canonical splice sites for minor introns.

This is corrected (line 148).

**Recommendations for the authors:**

**Reviewer #2 (Recommendations For The Authors):**
Figures 1, 3, and 4: I suggest that authors add regression lines.

We added the regression lines with the “pgls” function from the R package “caper” (in Fig. 1, 3 and 4, and also in all other figures where we present correlations).

Figure 2: As previously mentioned, the terms "minor introns" and "major introns" are extremely confusing. I strongly suggest the authors use different naming conventions.

We changed the terminology:

minor introns -> minor-isoform introns

major introns -> major-isoform introns

Figure 5: Intron-exon boundaries and splice site annotations are shown at the bottom of B, C, and D but not A. I suggest removing the annotation beneath B for consistency and since A+C and B+D are aligned on the x-axis.

Corrected, it was a mistake.

Figure 7: The yellow dotted line is very challenging to see in A.

Corrected, the line has been widened.